# Lipocalin 2 mediates appetite suppression during pancreatic cancer cachexia

Brennan Olson[1,2], Xinxia Zhu[1], Mason A. Norgard[1], Peter R. Levasseur[1], John T. Butler[1,2], Abigail Buenafe[1], Kevin G. Burfeind[1,2], Katherine A. Michaelis [1,2], Katherine R. Pelz[3], Heike Mendez[3], Jared Edwards [1], Stephanie M. Krasnow [1], Aaron J. Grossberg [3,4,5] & Daniel L. Marks [1,3,6✉]

Lipocalin 2 (LCN2) was recently identified as an endogenous ligand of the type 4 melano-cortin receptor (MC4R), a critical regulator of appetite. However, it remains unknown if this molecule influences appetite during cancer cachexia, a devastating clinical entity character-ized by decreased nutrition and progressive wasting. We demonstrate that LCN2 is robustly upregulated in murine models of pancreatic cancer, its expression is associated with reduced food consumption, and *Lcn2* deletion is protective from cachexia-anorexia. Consistent with LCN2's proposed MC4R-dependent role in cancer-induced anorexia, pharmacologic MC4R antagonism mitigates cachexia-anorexia, while restoration of *Lcn2* expression in the bone marrow is sufficient in restoring the anorexia feature of cachexia. Finally, we observe that LCN2 levels correlate with fat and lean mass wasting and is associated with increased mortality in patients with pancreatic cancer. Taken together, these findings implicate LCN2 as a pathologic mediator of appetite suppression during pancreatic cancer cachexia.

[1] Papé Family Pediatric Research Institute, Oregon Health & Science University, Portland, OR, USA. [2] Medical Scientist Training Program, Oregon Health & Science University, Portland, OR, USA. [3] Brenden-Colson Center for Pancreatic Care, Oregon Health and & Science University, Portland, OR, USA. [4] Department of Radiation Medicine, Oregon Health & Science University, Portland, OR, USA. [5] Cancer Early Detection Advanced Research Center, Knight Cancer Institute, Oregon Health & Science University, Portland, OR, USA. [6] Knight Cancer Institute, Oregon Health & Science University, Portland, OR, USA. ✉email: marksd@ohsu.edu

Cachexia, or disease-associated wasting, is a metabolic state consisting of a paradoxical decrease in appetite, yet increase in basal metabolic rate[1–3]. This mismatch in caloric intake and expenditure imposes a significant energy imbalance, leading to excessive wasting, reduced quality of life, and decreased patient tolerance to therapy[4,5]. Since decreased nutrition is an essential component of cancer cachexia[6,7], improving cachexia-related anorexia remains an important aspect in treating this metabolic disorder. Indeed, nutritional interventions and pharmacologic improvement of appetite are emerging as promising treatment paradigms for patients with cachexia[8–10]. Therefore, identification of the pathways that influence nutritional deficits associated with cachexia could provide new therapeutic options for this otherwise debilitating metabolic disorder.

To this end, significant efforts are being made to identify the mechanism of appetite dysregulation during cancer-associated cachexia. Lipocalin 2 (LCN2), also known as neutrophil gelatinase-associated lipocalin, siderocalin, or 24p3, is a pleiotropic mediator of several inflammatory and metabolic processes that is secreted into circulation during a variety of diseases associated with cachexia[11], including cancer[12], chronic kidney disease[13], and heart failure[14]. Recently published data demonstrate LCN2 suppresses appetite through its binding to the melanocortin 4 receptor (MC4R) in the paraventricular nucleus (PVN) of the hypothalamus[15], a nucleus essential in regulating feeding behaviors and energy homeostasis[16]. These studies demonstrate a role for LCN2 in regulating appetite during non-pathologic states. However, whether the dysregulation of LCN2 produces meaningful changes in energy balance during cancer-associated cachexia is not known.

Here, we explore the relationship between LCN2 production, feeding behaviors, and tissue catabolism during pancreatic cancer cachexia. In five separate rodent models of pancreatic cancer cachexia, we demonstrate a large induction of circulating and central LCN2 levels that negatively correlate with food intake and muscle mass. Since the literature supports the notion that *Lcn2* tissue expression is disease-dependent[11], we identify the bone marrow and myeloid compartment, namely neutrophils, as the predominant source of LCN2 during pancreatic cancer cachexia. We show that the genetic deletion of *Lcn2* mitigates cachexia–anorexia and lean and fat mass loss, while restoration of LCN2 in the bone marrow compartment alone reestablishes the anorexia and muscle catabolism features of cachexia. Importantly, we detect LCN2 in the brain after restoring expression in the bone marrow compartment, demonstrating LCN2 readily crosses the blood–brain barrier (BBB). Although inflammation is sufficient to induce anorexia, the increased food intake during LCN2 blockade is independent of systemic inflammatory state and immunologic activation. Through pair-feeding studies, we show that the muscle-sparing effects of LCN2 blockade are attributable to increased food intake alone, not differential regulation of ubiquitin- and autophagy-related catabolic pathways. Analogous to our murine models, we present human data showing associations between rising LCN2 levels and neutrophil expansion, fat and lean mass wasting, and mortality during pancreatic cancer. Taken together, we present an immunometabolic mechanism of appetite regulation during cachexia in which LCN2 is derived from the bone marrow compartment, crosses the BBB, and binds to the MC4R in mediobasal hypothalamic neurons to mediate long-term anorexia and subsequent loss of lean and fat mass.

## Results

**LCN2 is upregulated in the circulation and CNS in multiple murine models of pancreatic cancer cachexia and correlates with anorexia and muscle loss.** Using five separate murine models of pancreatic ductal adenocarcinoma-associated cachexia, we measured daily food intake, skeletal and cardiac muscle mass, and circulating and central LCN2 levels. These models follow distinct daily food intake trajectories, total food consumption, and terminal skeletal and cardiac muscle masses over the course of disease (Fig. 1a–d). Similarly, the transcriptional regulation of catabolic and inflammatory genes associated with cachexia varied considerably amongst these five models of pancreatic cancer cachexia. Consistent with previous reports of pancreatic cancer cachexia, skeletal muscle catabolism was associated with elevated expression of E3 ubiquitin ligase-related genes, including *Mafbx, Murf1, and Foxo1*, while cardiac atrophy was associated with increased autophagy-related genes, including *Bnip3, Ctsl*, and *Gabarapl* (Fig. 1e, f)[17]. Other features of cachexia, including hypothalamic inflammation and induction of acute-phase proteins by the liver, varied amongst these five models of pancreatic cancer cachexia. We observed significantly higher hypothalamic *Il1b* mRNA in two models (KPC and FC1245), while *Il1r1* was elevated in all models but FC1242 (Fig. 1g). The liver contributes to the overall systemic inflammatory phenotype during cachexia through production of acute-phase proteins and pro-inflammatory signals; acute-phase proteins *Apcs* and *Orm1* were transcriptionally elevated in all models, while variable increases in *Il1-β, Il-6,* and *Selp* mRNA were observed amongst models. Despite the varying degrees of cachexia symptoms (anorexia and muscle catabolism), as well as inflammatory and catabolic gene signatures, we observed increased LCN2 protein in the circulation and cerebrospinal fluid (CSF) in all models of pancreatic cancer cachexia (Fig. 1i, j). Furthermore, we observed a negative correlation between both peripheral and central LCN2 levels and food intake, as well as skeletal muscle mass, when comparing all models (Fig. 1k–n and Supplementary Fig. 1).

**LCN2 is predominantly produced in the bone marrow compartment and neutrophils during pancreatic cancer cachexia.** To determine the meaningful source of LCN2 during pancreatic cancer cachexia, we analyzed LCN2 levels across tissues utilizing the KPC model described in Fig. 1. During both normal physiology and cachexia, we observed LCN2 principally in the bone marrow compartment, with much smaller quantities observed in the spleen, lung, and liver (Fig. 2a). We observed significantly higher LCN2 protein levels in the lung, liver, and spleen during pancreatic cancer cachexia. However, we did not observe an increase in bone marrow LCN2 during pancreatic cancer cachexia (Fig. 2a, b), potentially owing to a saturating level in both sham and cachexia conditions. Expression of LCN2 in the bone marrow compartment was confirmed by immunohistochemical staining (Fig. 2c). Importantly, we validated the specificity of the LCN2 antibodies used in our studies, as staining is absent in *Lcn2-KO* mouse tissue, including the bone marrow (Supplementary Fig. 2A). The induction of LCN2 has been attributed to several canonical inflammatory cytokines, including IFNγ[18], TNFα[18], IL-1β[19], and IL-6[20] to name a few. To determine if LCN2 is an inflammation-induced mediator of cachexia, we examined circulating LCN2 levels in both *Il-6* and *Myd88* knockout mice, as these two inflammatory signaling pathways are critical in the development of several features of cachexia[6,21]. Indeed, circulating levels of LCN2 are significantly reduced in both *Myd88* and *Il-6* knockout tumor-bearing mice (Supplementary Fig. 2B, C). Since the bone marrow appeared to be the largest producer of LCN2 and is a major primary lymphoid organ responsible for immunologic response during disease, we hypothesized that circulating and tissue-resident immune cells were responsible, in part, for the observed increase in LCN2. To address this question, we analyzed the composition of circulating immune cells during

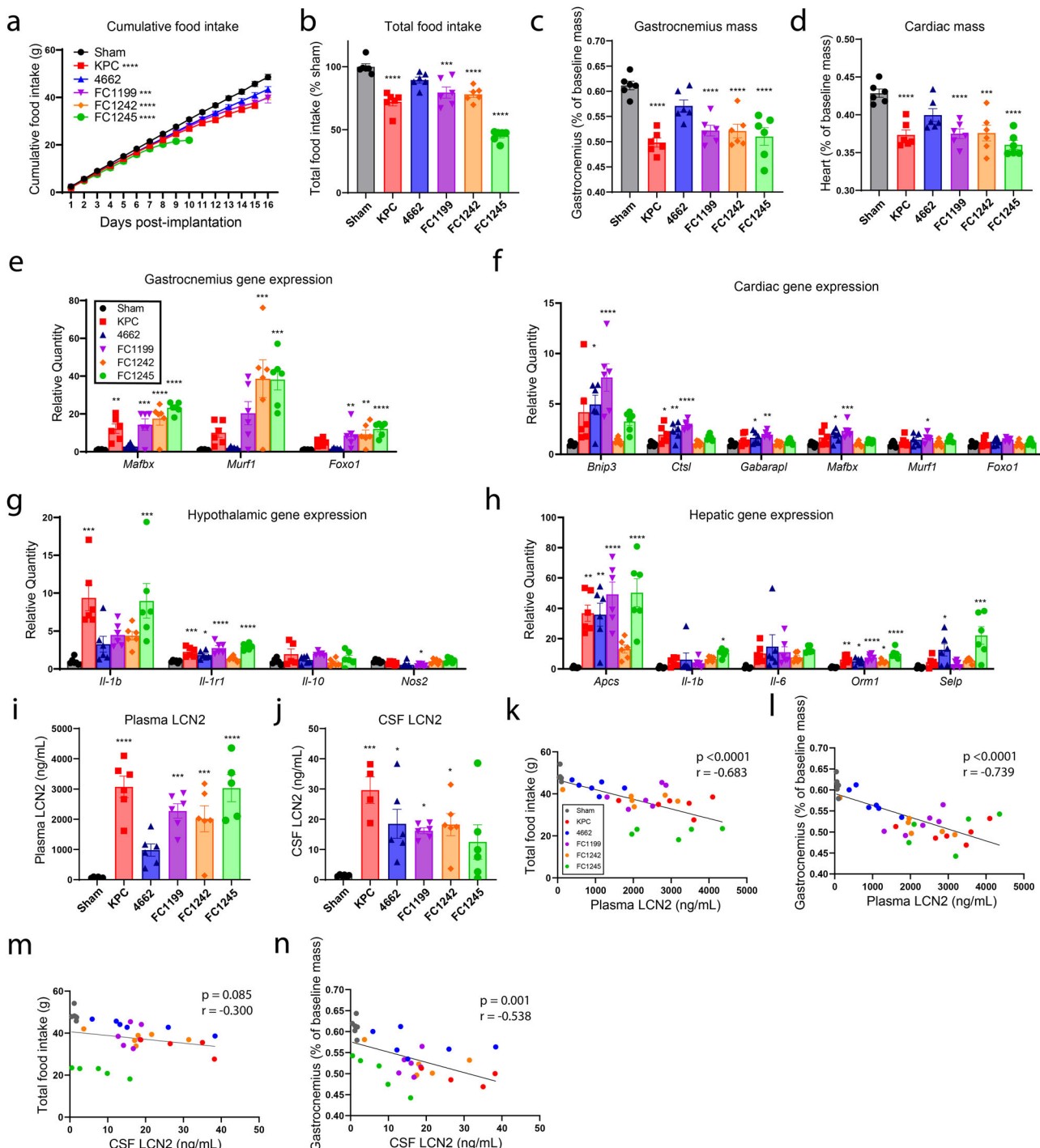

**Fig. 1 LCN2 is upregulated across several models of pancreatic cancer cachexia and correlates with food consumption and muscle loss. a** Cumulative food intake in five models of pancreatic cancer cachexia and sham operation controls. **b** Total food intake normalized to sham control group. **c** Skeletal muscle catabolism as indicated by terminal gastrocnemius mass normalized to body mass. **d** Cardiac muscle catabolism as indicated by terminal heart mass normalized to body mass. **e** Gastrocnemius, **f** heart, **g** hypothalamus, and **h** liver gene expression profiles in pancreatic cancer cachexia models (represented as relative quantity to sham control). Terminal **i** plasma and **j** CSF LCN2 levels. Linear regression analysis between plasma LCN2 levels and **k** total food intake or **l** gastrocnemius mass. Linear regression analysis between CSF LCN2 levels and **m** total food intake or **n** gastrocnemius mass. **a**–**i**, **k**, **l** $N = 6$ per group. **j**, **m**, **n** $N = 6$ per group except KPC ($N = 4$ per group). All data are expressed as mean ± SEM. Data represented in **b**–**j** were analyzed with one-way ANOVA with Bonferroni multiple comparisons comparing tumor experimental groups to sham operation control. **k**–**n** Analyzed by simple linear regression and two-tailed correlation analyses. *$p \leq 0.05$, **$p \leq 0.01$, and ***$p \leq 0.001$, ****$p \leq 0.0001$. Sham operation controls = gray/black, KPC = red, 4662 = blue, FC1199 = purple, FC1242 = orange, FC1245 = green.

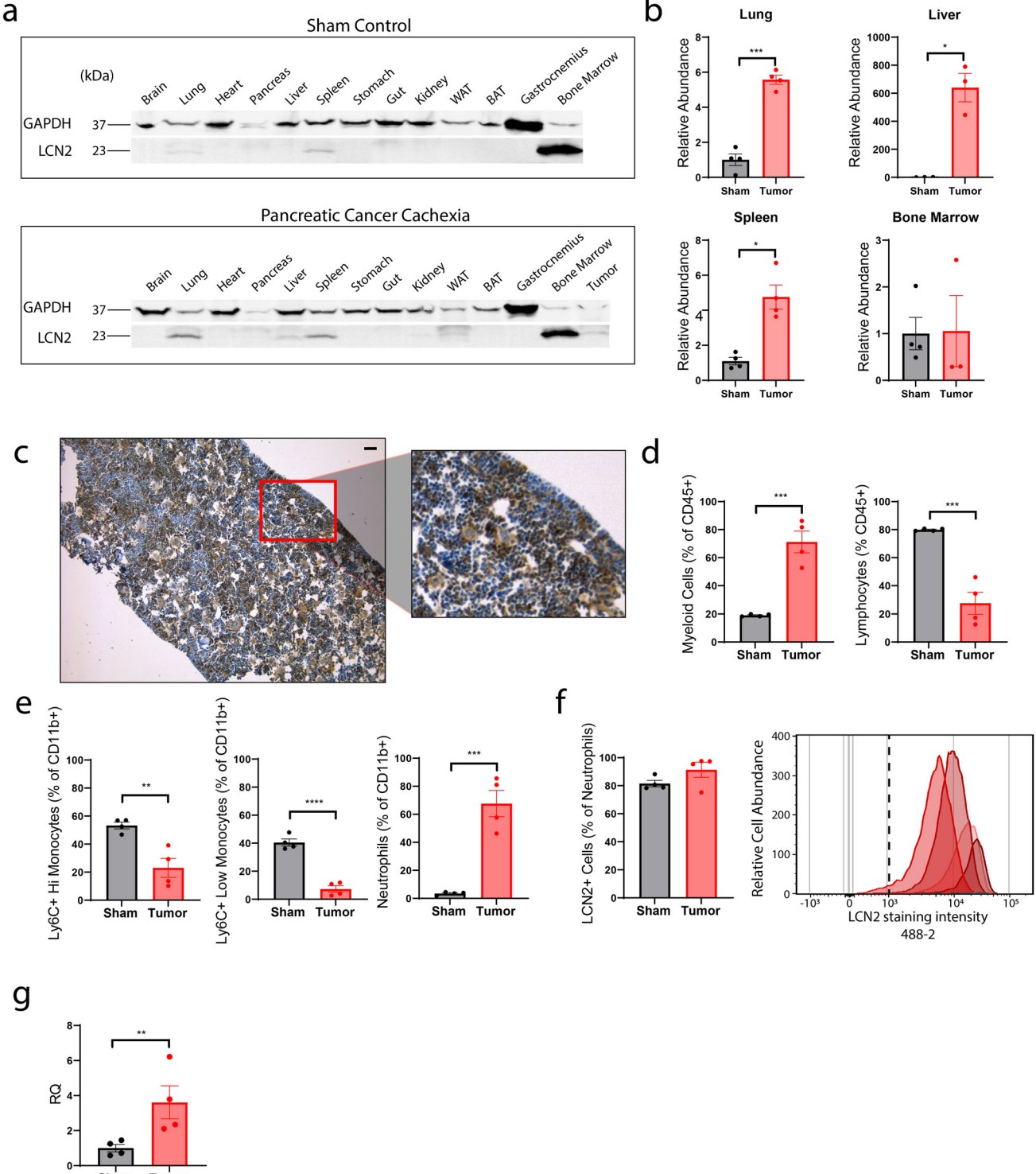

**Fig. 2 LCN2 is predominantly produced by the bone marrow compartment and neutrophils during cachexia. a** Representative tissue Western blot analysis of LCN2 in sham operated or pancreatic cancer cachexia mice (orthotopic KPC pancreatic cancer cell implantation). **b** Western blot quantification of LCN2 in lung, liver, spleen, and bone marrow compartments ($n = 4$ per group, except in KPC-engrafted liver and bone marrow samples [$n = 3$ per group]). **c** Representative immunohistochemistry staining images of LCN2 in the bone marrow of cachectic mice; 10× and high-resolution inset (scale bar = 50 μm). **d**, **e** Flow cytometry analysis to detect myeloid and lymphoid populations and myeloid cell subpopulations. **f** Flow cytometry quantification of intracellular LCN2 in Ly6G+ neutrophils; accompanying fluorescent intensity histogram, capturing relative abundance of Ly6G+ LCN2+ neutrophils between sham (gray lines) and cachectic (red lines) groups. The Ly6G+ cell population in the sham group is suppressed to the $X$-axis by the relative abundance found in the cachectic group. Each curve represents a single subject. Vertical dotted line approximates gating threshold for positive intracellular LCN2 staining. **g** FACS-sorted neutrophil RNASeq analysis of LCN2 transcripts. **d**–**g** $N = 4$ per group. **b**, **d**–**g** Analyzed by two-tailed Students $t$ tests. **a** LCN2 molecular weight = 23 kDa, GAPDH = 37 kDa. All data expressed as mean ± SEM. Quantitative data analyzed by Student's $t$ test. *$p \leq 0.05$, **$p \leq 0.01$, ***$p \leq 0.001$, and ****$p \leq 0.0001$. Sham operation controls = gray/black, KPC = red.

cachexia. We observed a large expansion of the myeloid cell population, with a relative and absolute decrease in the lymphocyte population (Fig. 2d). This myeloid expansion in murine pancreatic cancer cachexia is composed principally of circulating neutrophils, with relative decreases in Ly6C+ high and low expressing monocytes (Fig. 2e and Supplementary Fig. 2D, E). Consistent with previous reports[22], nearly all neutrophils stain positive for LCN2 during both normal physiology and cachexia (Fig. 2f). However, we detected a significant increase in *Lcn2* transcripts in circulating neutrophils of cachectic mice by RNA sequencing (Fig. 2g).

**Genetic deletion of Lcn2 ameliorates pancreatic cancer cachexia–anorexia**. After observing upregulation of LCN2 in the periphery and CNS of mice that develop pancreatic cancer cachexia, we wanted to determine if genetic deletion of *Lcn2* would reverse the behavioral and physiologic manifestations of cachexia. After KPC tumor implantation, we observed increased food consumption in *Lcn2-KO* mice relative to wild-type (WT) control (Fig. 3a, b). Furthermore, we observed a statistically nonsignificant increase in skeletal muscle mass in tumor-bearing *Lcn2-KO* mice compared to WT, while cardiac tissue was significantly spared in tumor-bearing *Lcn2-KO* mice (Fig. 3c, d). Similar to the tissue-specific measurements, nuclear magnetic resonance (NMR) body composition analysis of the lean compartment also demonstrates a nonsignificant improvement in lean mass of *Lcn2-KO* tumor-bearing mice (Fig. 3e). NMR-based body composition analysis of the fat compartment demonstrated consistent increase in fat mass of *Lcn2-KO* tumor-bearing mice throughout the study, including early (day 4 post implantation), mid (day 8 post implantation), and late-stage (day 11 post implantation) cachexia for *Lcn2-KO* tumor-bearing mice (Fig. 3f–h). Similarly, *Lcn2-KO* tumor-bearing mice had a significantly increased inguinal fat pad mass at the end of the study when normalizing to sham controls (Fig. 3i). Despite observing an elevation in fat mass of *Lcn2-KO* tumor-bearing mice, there was no difference in white-adipose browning between WT and *Lcn2-KO* mice as indicated by browning genes *Ucp1, Prdm16*, and *Cidea* (Supplementary Fig. 3A–D). Although there was a significant increase in *Ppar-γ* of *Lcn2-KO* mice, controlling for baseline genotypic expression did not result in a significant reduction in tumor-bearing mice between genotypes (Supplementary Fig. 3C). Histologically, there appears to be an increase in cellularity in WT tumor-bearing mice compared to *Lcn2-KO* tumor-bearing mice, suggestive of increased cellular infiltrates known to be important in fat wasting in viral cachexia models (Supplementary Fig. 3E)[23]. While this particular model of cancer cachexia results in significant lean and fat mass loss, tumor-free body mass is unchanged owing to abdominal ascites and third-spacing edema (Supplementary Fig. 3F)[17].

*Lcn2* deletion improved voluntary wheel running and trended toward improved survival duration (Supplementary Fig. 3G–J). These behavioral effects were independent of tumor growth and basic histologic appearance, as tumor mass and key histopathological tumor characteristics (including ragged parenchymal infiltrations, focal necrotic loci, and acute inflammation) were similar between WT and *Lcn2-KO* mice (Supplementary Fig. 3K–N). The impacts on feeding behavior and voluntary wheel running were also observed in animals with a less precipitous disease course (by orthotopic implantation of fewer pancreatic cancer cells) (Supplementary Fig. 3O, P). Since it is suggested that gastrointestinal malabsorption may play a role in energy balance during cancer cachexia, we analyzed fecal lipid, protein, and protease concentrations and did not detect changes during cachexia nor genotype-specific alterations, except for an increase in fecal lipids of *Lcn2-KO* mice at the final 5 days of the study (days 7–11), although this may be due to increased food intake as these animals were not pair-fed (Supplementary Fig. 3Q–U). We also monitored blood glucose of WT and *Lcn2-KO* mice and observed a significant increase in *Lcn2-KO* tumor-bearing mice compared to WT at the end of the study, likely due to the increased food consumption of *Lcn2-KO* tumor-bearing mice, as WT and *Lcn2-KO* sham controls remained similar throughout the study (Supplementary Fig. 3V–X). We did not detect LCN2 in the circulation or CSF of tumor-bearing *Lcn2-KO* mice, demonstrating that tumor-derived LCN2 does not meaningfully contribute to circulating or central LCN2 levels during cachexia (Fig. 3j, k). Furthermore, genetic deletion of *Lcn2* did not affect the myeloid to lymphocyte ratio during pancreatic cancer cachexia, demonstrating a similar immunologic profile amongst *Lcn2-KO* and WT genotypes (Fig. 3l, m). Despite improvements in skeletal and cardiac tissue mass, deletion of *Lcn2* did not alter ubiquitin ligase or autophagy-related transcripts during cachexia (Fig. 3n, o). Finally, *Lcn2* deletion did not alter acute-phase- and inflammatory-related transcripts in the liver and hypothalamus during the development of cachexia (Fig. 3p, q).

**The central melanocortin 4 system mediates appetite suppression during pancreatic cancer cachexia**. Based on recent literature demonstrating that LCN2 binds to hypothalamic MC4R[15] and our observation that feeding behavior improves in *Lcn2-KO* mice during cachexia, we hypothesized that MC4R antagonism (or inverse agonism) would improve appetite during the development of cancer cachexia. To address this hypothesis, we examined whether intracerebroventricular (ICV) infusion of agouti-related peptide (AgRP), an inverse agonist of the MC4R, improved appetite during the progression of pancreatic cancer cachexia. After the development of the signs and symptoms of cachexia (lethargy, decreased food intake, minimal grooming), we began ICV administration of AgRP (Fig. 4a). Mice consumed equivalent amounts of food prior to ICV AgRP treatment (Fig. 4b). However, after the development of cachexia symptoms, mice receiving ICV AgRP treatment increased food intake compared to their vehicle-treated counterparts (Fig. 4c, d). Similar to the *Lcn2-KO* tumor-bearing mice, AgRP-treated mice had improved skeletal and cardiac tissue mass at the end of the study (Fig. 4e, f). Furthermore, these muscle-sparing effects were independent of ubiquitin-proteasome and autophagy-related pathways, suggesting the increased food consumption alone was responsible for the sparing of muscle mass (Supplementary Fig. 4A, B). Tumor burden was equivalent between AgRP- and vehicle-treated groups (Supplementary Fig. 4C).

**LCN2 crosses the BBB and regulates feeding behaviors during cancer cachexia**. Since we observed that the bone marrow is the predominant location of LCN2 protein production, we hypothesized that restoring *Lcn2* expression in this compartment would result in significant production and secretion of LCN2 into circulation, and that this circulating LCN2 is capable of crossing the BBB to regulate appetite through its recently identified actions in the CNS. Using an adoptive transfer technique that gently conditions the bone marrow while sparing the integrity of the BBB, we transplanted WT bone marrow into *Lcn2-KO* mice (Fig. 5a). Indeed, restoration of *Lcn2* in the bone marrow compartment alone resulted in a significant rescue of peripheral LCN2 levels during both normal physiology (sham operation mice) and cancer cachexia (Fig. 5b). Furthermore, restoration of *Lcn2* expression in the bone marrow yielded a near 50% restoration of LCN2 levels in the CSF during both normal

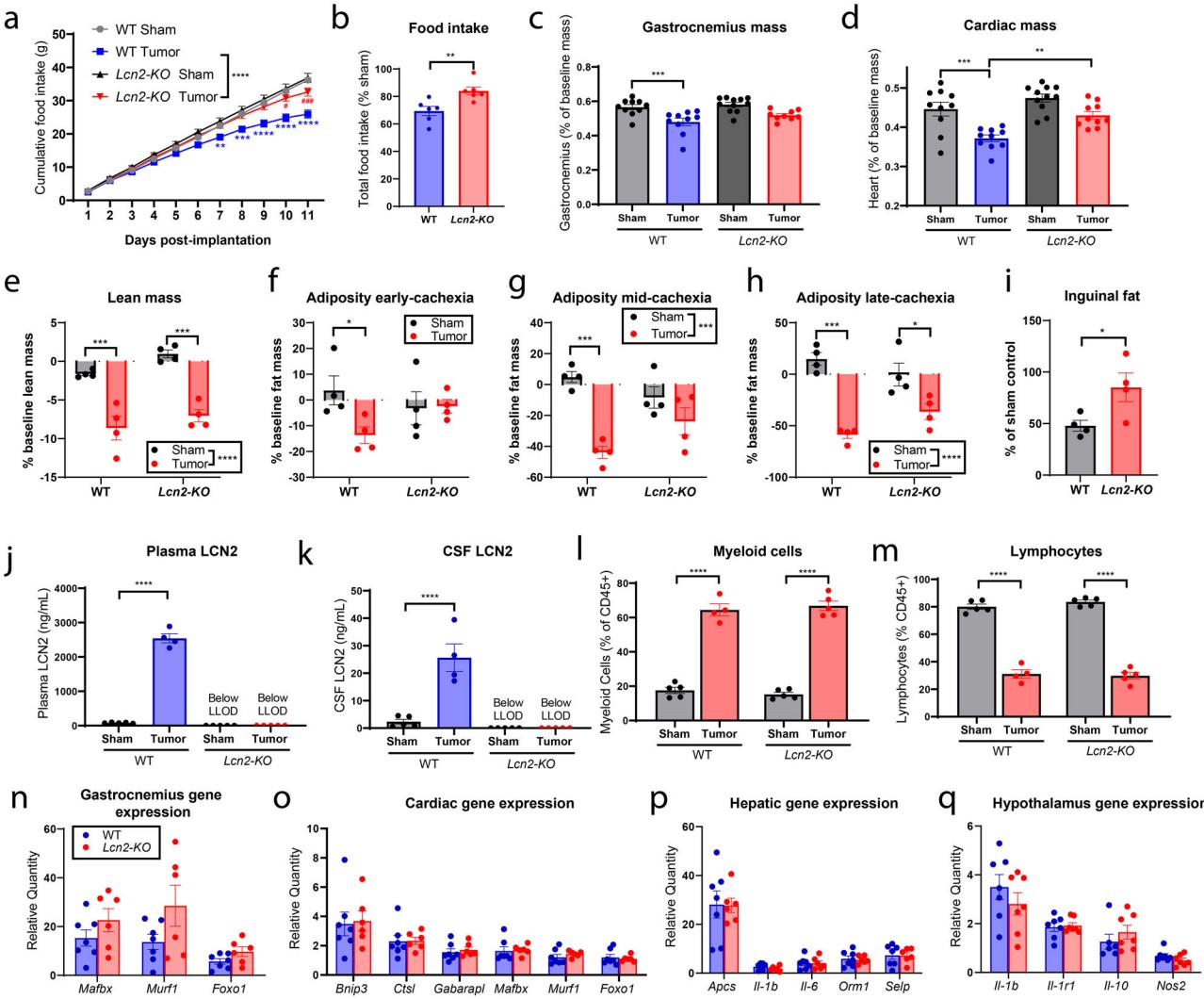

**Fig. 3 Genetic deletion of LCN2 ameliorates cachexia–anorexia. a** Cumulative and **b** total food intake for WT and *Lcn2-KO* mice after receiving tumor implantations or sham operations ($n = 6$ per group for WT sham, WT tumor, and *Lcn2-KO* tumor groups; $n = 7$ for *Lcn2-KO* sham controls). **c** Terminal gastrocnemius and **d** heart mass as a percentage of baseline body mass ($n = 10$ per group in WT sham, WT tumor, and *Lcn2-KO* tumor groups; $n = 11$ in *Lcn2-KO* sham controls). **e** NMR body composition analysis of terminal tumor-free lean ($n = 4$ per group). **f–h** NMR body composition analysis of fat mass at early (study day 4), mid (study day 8), and late (study day 11) cachexia ($n = 4$ per group). **i** Terminal inguinal fat pad mass normalized to genotype control ($n = 4$ per group). **j** Terminal plasma and **k** cerebrospinal fluid LCN2 levels ($n = 5$ per group for WT Sham, *Lcn2-KO* sham, and *Lcn2-KO* tumor groups; $n = 4$ for the WT tumor group). **l** Proportions of myeloid and **m** lymphoid cells as a percentage of CD45+ cells ($n = 5$ per group for WT Sham, *Lcn2-KO* sham, and *Lcn2-KO* tumor groups; $n = 4$ for the WT tumor group). **n** Ubiquitin proteasome pathway gene expression in the gastrocnemius. **o** Ubiquitin proteasome and autophagy-related gene expression in cardiac muscle. **p** Hepatic expression of acute-phase- and inflammatory-related transcripts. **q** Hypothalamic gene expression of inflammation-related transcripts. For **n–q**, $n = 6$ per group for WT sham and *Lcn2-KO* tumor groups; $n = 7$ for *Lcn2-KO* sham and WT tumor, groups. All gene expression data represented as fold change over genotype-matched controls. No difference was observed in baseline expression of transcripts between WT and *Lcn2-KO* mice. All data expressed as mean ± SEM. Cumulative food intake data were analyzed by a repeated-measures Two-way ANOVA followed by Bonferroni's post hoc test. Food intake and inguinal fat mass as % of sham was analyzed by two-tailed Student's *t* test. All other data (**c–h**, **j–q**) were analyzed by ordinary Two-way ANOVA followed by Bonferroni's post hoc test. *$p \leq 0.05$, **$p \leq 0.01$, ***$p \leq 0.001$, and ****$p \leq 0.0001$. LLOD lower limit of detection. **a–d**, **j**, **k**, **n–q** Gray = WT sham operation control; black = *Lcn2-KO* sham operation control; blue = WT KPC-engrafted mice; red = *Lcn2-KO* KPC-engrafted mice. **e–i**, **l**, **m** Sham operation controls = gray/black, KPC-engrafted mice = red.

physiology and cachexia (Fig. 5c). We then tested whether cerebral LCN2 is sufficient to alter feeding behavior. Consistent with previous reports, we demonstrated that chronic ICV injection of LCN2 is capable of inducing appetite suppression and concurrent weight loss (Supplementary Fig. 5A–E)[15]. Furthermore, we observed elevated cFos expression in paraventricular neurons following ICV administration of LCN2; this cFos expression profile is similar to that of mice treated with ICV Melanotan-II (MT-II), a potent synthetic analog of α-MSH and agonist of the

MC4R (Supplementary Fig. 5F, G). To ensure LCN2's anorectic response is dependent on MC4R signaling, we repeatedly administered LCN2 to the lateral ventricle of WT and *Mc4r*-KO mice, and observed a significant reduction in food intake, body mass, and fat mass in WT compared to *Mc4r*-KO mice (Supplementary Fig. 5H–L).

**Restoration of Lcn2 in the bone marrow compartment suppresses food intake during pancreatic cancer cachexia.** To

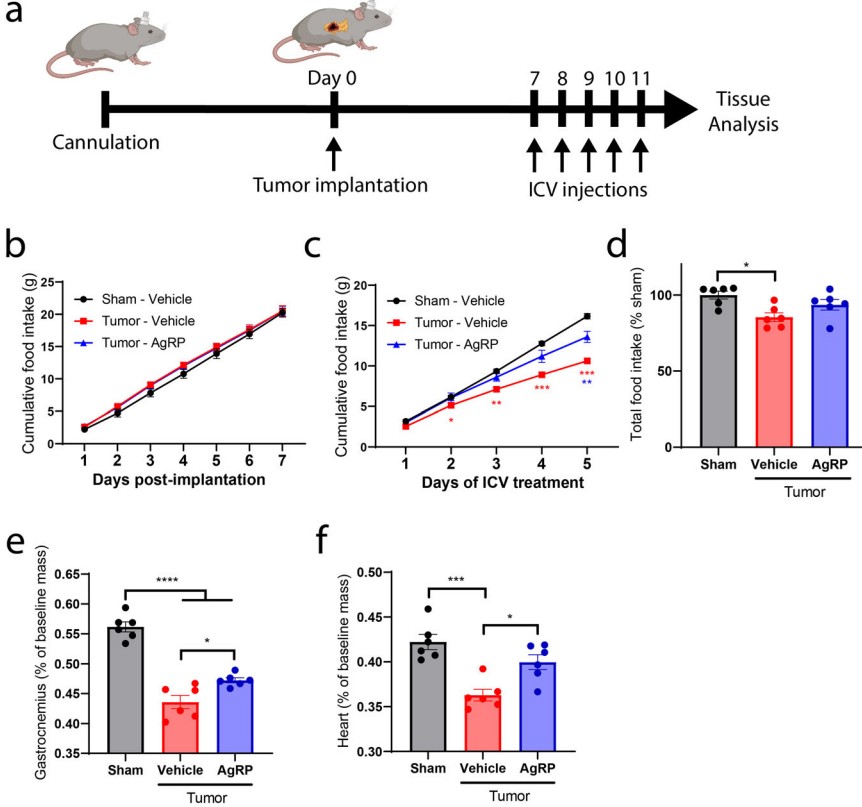

**Fig. 4 MC4R inverse agonism improves feeding behaviors and muscle mass during pancreatic cancer cachexia. a** Experimental design and protocol of cannulation, tumor implantation, and ICV treatment. Cumulative food intake after tumor implantation **b** before and **c** after initiation of daily ICV AgRP (1 nmol) or vehicle treatments. In **b**, the Tumor–AgRP group is masked immediately behind the Tumor–Vehicle group. **d** Total food intake as a percentage of sham control for entire study. **f** Terminal gastrocnemius and **e** cardiac tissue mass normalized to baseline mass. $n = 6$ per group. All data expressed as mean ± SEM. Cumulative food intake data were analyzed by a repeated-measures Two-way ANOVA followed by Bonferroni's post hoc test. All other data were analyzed by ordinary One-way ANOVA followed by Bonferroni's post hoc test. *$p ≤ 0.05$, **$p ≤ 0.01$, ***$p ≤ 0.001$, and ****$p ≤ 0.0001$. Black/gray = sham operation control, red = KPC-engrafted, ICV vehicle treated mice, blue = KPC-engrafted, ICV AgRP-treated mice.

examine whether restoration of *Lcn2* expression in the bone marrow compartment influences energy balance during pancreatic cancer cachexia, we engrafted WT bone marrow into *Lcn2-KO* mice, allowed transplanted animals to recover for 8 weeks, then orthotopically implanted pancreatic cancer cells and monitored feeding behavior. Using this bone marrow transplantation (BMT) method, we regularly achieve high levels of chimerism (>80%) as indicated by GFP expression in circulating immune cells in both genotypes (WT and *Lcn2-KO*) and experimental groups (sham operation and tumor implantation) (Supplementary Fig. 6A–D). After BMT and subsequent tumor implantation, we observed a decrease in daily cumulative food intake and a trend toward decreased total food intake for *Lcn2-KO* mice receiving *Lcn2*-replete bone marrow (Fig. 5d, e). As we observed previously, there was a modest improvement in skeletal and cardiac muscle catabolism in the pure *Lcn2-KO* genotype (*Lcn2-KO* mice receiving *Lcn2-KO* bone marrow) compared to the WT genotype (WT mice receiving WT bone marrow) (Fig. 5f, g). In *Lcn2-KO* mice receiving WT bone marrow, we observed an intermediate skeletal and cardiac muscle wasting phenotype compared to the pure WT (WT mice receiving WT marrow) and *Lcn2-KO* (*Lcn2-KO* mice receiving *Lcn2-KO* marrow) genotypes. Successful bone marrow engraftment and expression of *Lcn2* were validated through Western blot analysis of the bone marrow at the end of the study (Supplementary Fig. 6C). Finally, we did not observe a difference in tumor burden, as indicated by terminal tumor mass, amongst the three experimental groups (Supplementary Fig. 6E).

**Lean mass sparing effect of Lcn2 ablation is mediated by increased food intake**. To formally test whether the muscle-sparing effects of *Lcn2* ablation are mediated through food intake, rather than a combination of food intake and tissue-specific metabolic alterations, we performed pair-feeding studies in which we matched the tumor-bearing *Lcn2-KO* food intake to that of their tumor-bearing WT counterparts (Fig. 6a). After pair feeding, we observed no difference in skeletal or cardiac tissue catabolism between *Lcn2-KO* and WT tumor-bearing mice (Fig. 6b, c). Collectively, this observation supports the hypothesis that the muscle-sparing effects of LCN2 are largely mediated through increased energy consumption, but not a direct effect of LCN2 on muscle.

**Circulating LCN2 correlates with neutrophil expansion, lean and fat mass wasting, and mortality in patients with pancreatic cancer**. To test whether our observations in these preclinical models of cachexia extend to human disease, we measured plasma LCN2 levels in patients with pancreatic cancer at diagnosis and, in several cases, follow-up visits. Since we observed an increase in circulating neutrophils, as well as their transcriptional upregulation of *Lcn2* in our rodent models of pancreatic cancer cachexia, we hypothesized that neutrophil expansion would be associated with an increase in circulating LCN2 levels during human disease. Indeed, we observed a concurrent increase in peripheral neutrophils, decrease in lymphocytes, and elevation of neutrophil-to-lymphocyte ratio (NLR) as circulating LCN2 levels rise, recapitulating the immunologic shift we observed in our

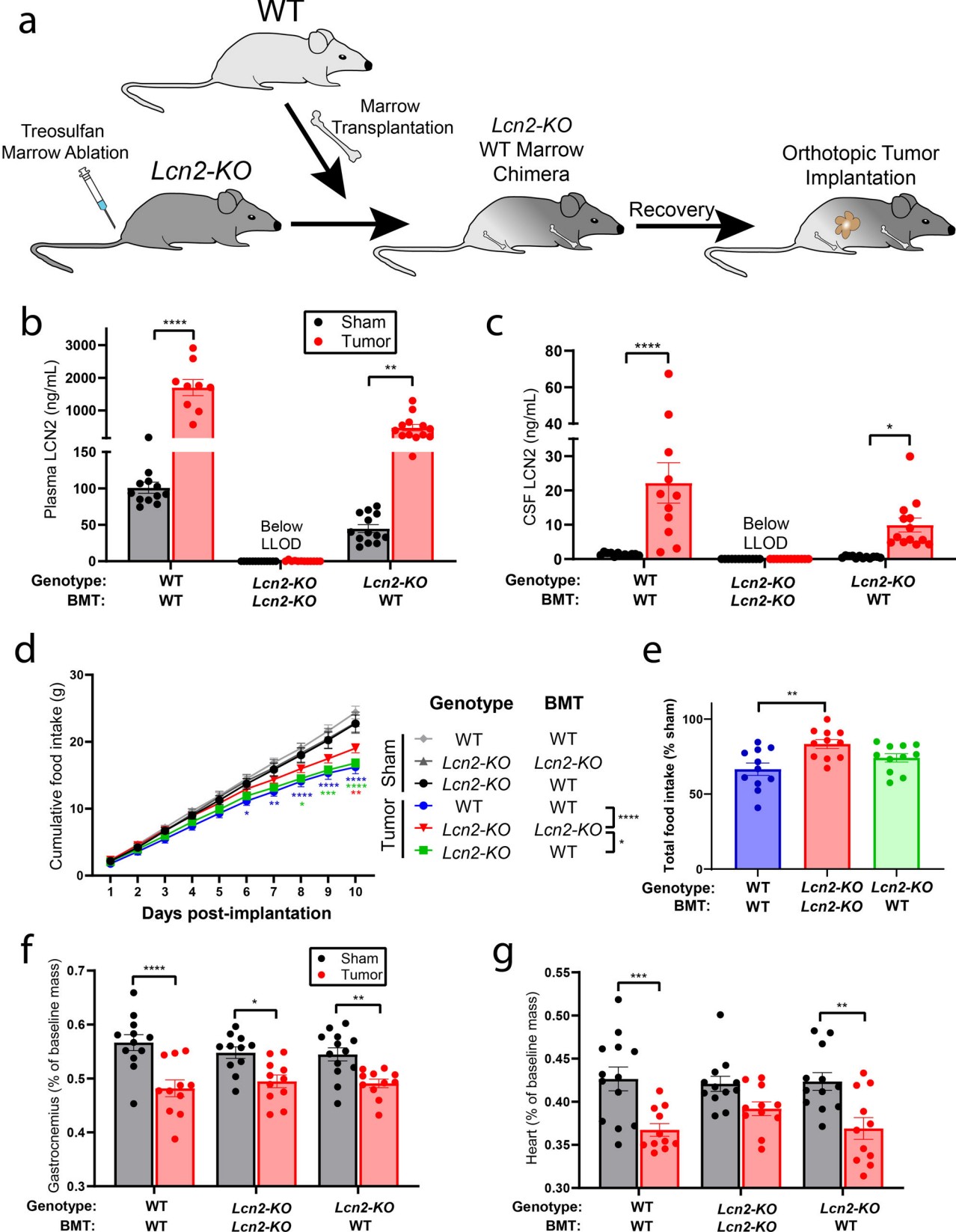

rodent models (Fig. 7a–c and Supplementary Table 1). Next, we sought to determine if an increase in LCN2 in patients diagnosed with pancreatic cancer corresponded with alterations in fat and lean mass, as these were consistent findings in our rodent models. We measured the amount of skeletal muscle and visceral adipose tissue in axial images at the third lumbar vertebrae for patients with pancreatic cancer at diagnosis, as well as follow-up visits (Fig. 7d)[24]. We observed a significant association between increasing LCN2 levels and loss of both visceral fat and skeletal muscle (Fig. 7e, f and Supplementary Table 2). Finally, we

**Fig. 5 LCN2 readily crosses the BBB and its expression in the bone marrow is sufficient to induce appetite suppression during pancreatic cancer cachexia. a** Experimental design of bone marrow transplantation experiments. **b** Terminal peripheral and **c** central LCN2 levels. **d** Cumulative food intake and **e** total food intake as a percent of sham control. **f** Terminal gastrocnemius and **g** cardiac tissue mass normalized to baseline mass. **b, c** $N = 12$ for WT/WT-BM sham, *Lcn2-KO*/KO-BMT sham, and *Lcn2-KO*/KO-BMT KPC-engrafted mice; $N = 9$ for WT/WT-BM tumor group; $N = 13$ for *Lcn2-KO*/WT-BMT sham and *Lcn2-KO*/WT-BMT KPC-engrafted mice. **d–g** $N = 12$ per group for WT/WT-BM sham and *Lcn2-KO*/KO-BMT sham groups; $N = 13$ for the *Lcn2-KO*/WT-BMT sham group; $N = 11$ per group for WT/WT-BM tumor, *Lcn2-KO*/WT-BMT tumor, and *Lcn2-KO*/KO-BMT tumor groups. All data expressed as mean ± SEM. Cumulative food intake data were analyzed by Mixed Model ANOVA with continuous measures followed by Bonferroni's post hoc test. Total food intake data were analyzed by One-way ANOVA followed by Bonferroni's post hoc test. All other data were analyzed by Mixed Model ANOVA followed by Bonferroni's post hoc test. Main column effects were calculated amongst tumor groups in **d** and are represented in the figure legend. *$p \leq 0.05$, **$p \leq 0.01$, ***$p \leq 0.001$, and ****$p \leq 0.0001$. LLOD lower limit of detection. BMT bone marrow transplantation. **b, c, f, g** Sham operation controls = gray/black, KPC-engrafted mice = red. **d, e** Light gray = WT/WT-BMT sham operation control, dark gray = *Lcn2-KO*/*Lcn2-KO*-BMT sham operation control, black = *Lcn2-KO*/WT-BMT sham operation control, blue = WT/WT-BMT KPC-engrafted mice, red = *Lcn2-KO*/*Lcn2-KO*-BMT KPC-engrafted mice, green = *Lcn2-KO*/WT-BMT KPC-engrafted mice.

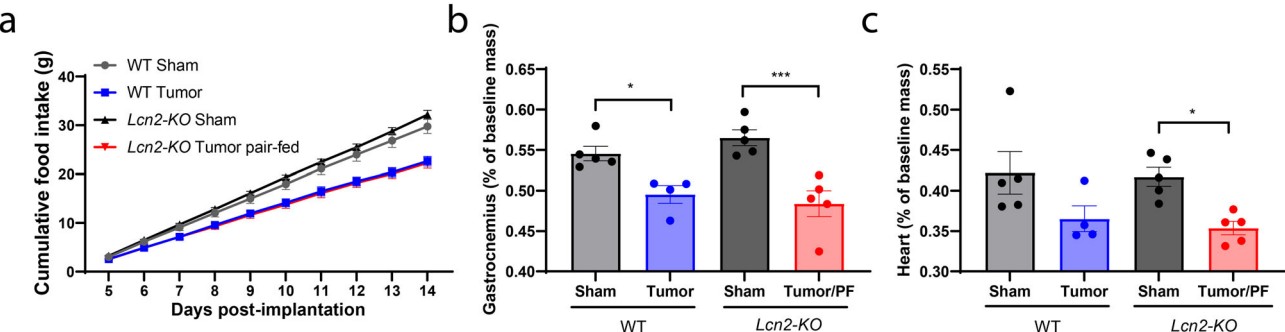

**Fig. 6 Pair feeding abolishes muscle gain in *Lcn2-KO* mice during cachexia. a** Cumulative food intake after the initiation of cachexia symptoms. $N = 5$ per group. **b** Terminal gastrocnemius and **c** cardiac tissue mass normalized to baseline mass. $N = 5$ per group for WT sham, *Lcn2-KO* sham, and *Lcn2-KO* tumor groups; $N = 4$ for the WT tumor group. Cumulative food intake data were analyzed by a repeated-measures Two-way ANOVA followed by Bonferroni's post hoc test. Data presented in **b, c** were analyzed by a Two-way ANOVA followed by Bonferroni's post hoc test. *$p \leq 0.05$, **$p \leq 0.01$, ***$p \leq 0.001$, and ****$p \leq 0.0001$. PF pair-fed. All data are expressed as mean ± SEM. Gray = WT sham operation control; black = *Lcn2-KO* sham operation control; blue = WT KPC-engrafted mice; red = *Lcn2-KO* KPC-engrafted, pair-fed mice.

identified a 240-ng/mL plasma LCN2 cutoff that stratifies patients by survival outcomes on univariate analysis[25]. In this patient population, a plasma LCN2 level of >240 ng/mL at diagnosis was associated with significantly decreased overall survival compared to patients with LCN2 values under this threshold (Fig. 7g and Supplementary Table 3: HR = 1.81; 95% CI = 1.07–3.05; $p = 0.03$). Taken together, our data demonstrate that elevated LCN2 levels during human pancreatic cancer are associated with neutrophil expansion, increased fat and skeletal muscle catabolism, and decreased survival. In general, these findings recapitulate what we observed in our pancreatic cancer cachexia mouse models.

## Discussion

The purpose of this work was to evaluate the role of LCN2 in mediating the anorectic component of pancreatic cancer cachexia. Herein, we describe an immunometabolic pathway, in which bone marrow upregulation and secretion of LCN2 results in appetite suppression through its action in the CNS. This direct effect on food intake results in an accompanying decrease in fat and lean mass in our rodent models (Fig. 8). In humans, we observe a similar immunologic shift and upregulation of LCN2 during the progression of pancreatic cancer. Although caloric intake is not readily captured in the clinic, we observed that an increase in circulating LCN2 is associated with fat and lean mass loss by computational analysis of fat and lean tissue using computed tomographic scans. Collectively, these studies support the notion that the pathologic upregulation of LCN2 contributes to the development of cancer-associated anorexia and both fat and lean mass wasting.

With the recent identification of LCN2 as a neuroendocrine hormone that binds to MC4R under normal physiologic conditions, we sought to expand upon this axis in the context of appetite dysregulation during neoplastic disease[15]. It is well described that LCN2 is upregulated during infectious inflammatory conditions[26], and since systemic inflammation is a hallmark feature of cancer cachexia, we hypothesized that the underlying sterile inflammatory state would result in a sustained elevation of circulating LCN2. Our data show a consistent elevation of circulating and central LCN2 levels in five separate rodent models of pancreatic cancer that induce variable degrees of cachexia. This elevation in circulating and central LCN2 levels is negatively correlated with food consumption and lean muscle mass. After our initial characterization of LCN2 production across five separate models of cachexia, we focused on the well-characterized KPC model for ensuing genetic and pharmacologic studies. Since several inflammatory mediators are shown to regulate LCN2 levels in various pathologies, we demonstrate that mice devoid of *Il-6* and *Myd88*, two inflammatory pathways known to mediate host metabolism during cachexia[6,21], have reduced circulating LCN2 levels, demonstrating LCN2 is an inflammation-induced protein during cancer cachexia. We identified the bone marrow compartment as the largest source of LCN2 protein. As the bone marrow is a primary lymphoid organ responsible for the generation of immune cells, we went on to identify neutrophils as a significant source of LCN2 during cachexia. Notably, we also observed an increase in lung, liver, and splenic LCN2 production during pancreatic cancer progression, likely attributable to increased neutrophilic infiltration during cachexia[27] and other systemic inflammatory states[28]. Finally, RNA sequencing of

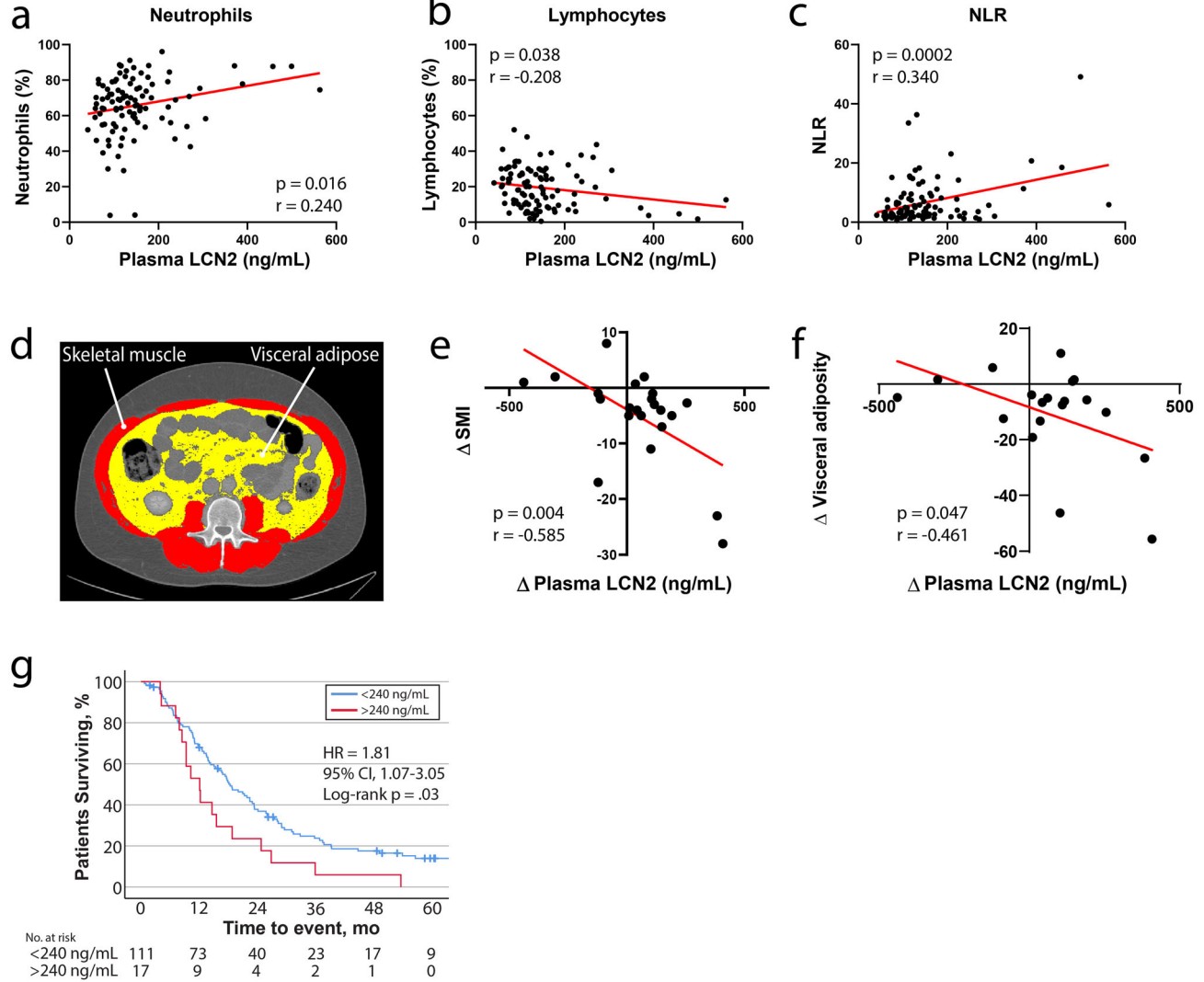

**Fig. 7 LCN2 is regulated during pancreatic cancer in humans and is associated with neutrophil expansion, skeletal muscle catabolism, and increased mortality.** Scatter plot of plasma LCN2 levels and **a** circulating neutrophil percentage, **b** lymphocyte percentage, and **c** neutrophil to lymphocyte ratio ($n = 100$). **d** Representative axial computed tomography scan at the third lumbar vertebrae highlighting visceral adipose tissue in yellow and skeletal muscle in red. **e** Correlation plot of change in skeletal muscle index and change in LCN2 levels after diagnosis ($n = 22$). **f** Correlation plot of change in visceral adiposity and change in LCN2 levels after diagnosis ($n = 19$). **g** Overall survival for patients with pancreatic cancer dichotomized by 240-ng/mL LCN2 levels at diagnosis ($n = 128$; two patients with <1.5 months of follow-up were excluded from analysis). SMI skeletal muscle index; both SMI and visceral adiposity calculated by dividing total cross-sectional area at L3 ($cm^2$) by patient height squared ($m^2$). Data from **e**, **f** represent a subset of patients from **a–c**, which have baseline and follow computed tomography scans. Please see Supplementary tables for demographics information for each patient population in **a–c**, **e–g**. **a–c**, **e**, **f** Analyzed by simple linear regression and two-tailed correlation analyses. Data represented in **g** analyzed by the log-rank Mantel-Cox test (two-sided). NLR neutrophil-to-lymphocyte ratio. **g** Blue = <240-ng/mL LCN2, red = >240-ng/mL LCN2.

circulating neutrophils of sham and tumor-bearing mice demonstrated a significant increase in *Lcn2* transcripts of cachectic mice. Taken together, the robust increase in neutrophil count, along with the modest elevation of intracellular LCN2 levels and increase in *Lcn2* expression in neutrophils, all support the notion that neutrophils are a significant source of circulating LCN2 levels during cachexia.

We next showed that genetic deletion of *Lcn2* significantly increased food consumption and modestly spared both lean and fat mass loss during pancreatic cancer cachexia. Notably, these appetite and tissue-sparing effects were independent of central and peripheral inflammatory status, as hypothalamic and hepatic inflammatory gene expression profiles were unchanged in mice with and without intact *Lcn2* expression. We also showed that the muscle-sparing effects of *Lcn2* deletion were independent of

ubiquitin-proteasome and autophagy-related pathways, two well-described catabolic pathways associated with lean mass wasting during chronic disease[29–33]. Furthermore, we did not detect significant alterations in white-adipose browning genes *Ucp1, Cidea, Ppar-γ,* and *Prdm16*. Recent studies have also concluded that *Lcn2* is dispensable in central and peripheral inflammatory response using LPS-based sepsis models[34]. However, opposing studies exist, and suggest LCN2 is a critical anti-inflammatory factor during sepsis[35], acute liver injury[36], and nephritic disease[37]. Thus, the role of LCN2 during disease is likely multifactorial and context-specific, with distinct physiologic roles depending on cellular source, underlying inflammatory insult, chronicity of disease, and molecular form of LCN2 (bound or unbound to a siderophore–iron complex). Furthermore, these disparate findings suggest that alternative epigenetic or post-

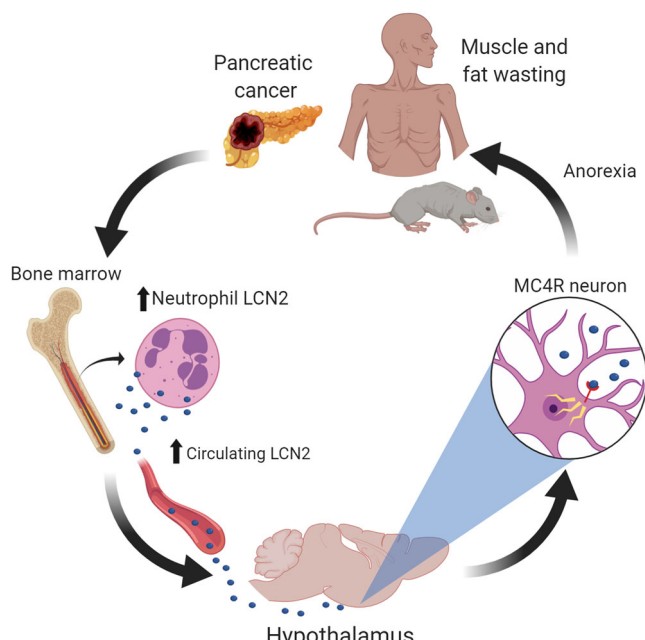

**Fig. 8 Graphical summary of findings.** We report that the pathogenesis of pancreatic cancer cachexia is associated with increased circulating levels of LCN2, which is an anorectic molecule that potentiates muscle and fat wasting associated with cachexia.

translational regulation could be involved in the regulation and function of LCN2 during states of metabolic stress. As it relates to the well-defined role of LCN2 in iron trafficking[26,38], Song et al. identified that holo-LCN2 (LCN2 bound to a bacterial siderophore–iron complex) promotes the generation of reactive oxygen species to impair oxidative phosphorylation in rat cardiomyocytes[39]. However, apo-LCN2 (LCN2 moiety alone) did not alter mitochondrial processes, demonstrating distinct molecular roles for the two forms of LCN2[39]. Future studies should account for apo- and holo-forms of LCN2 when discussing biological significance during disease. As it relates to the present study, it is likely that LCN2 exists almost exclusively in the apo-form in our cachexia models, as we have no evidence of bacterial proliferation during disease progression.

There is support for appetite and macromolecule regulating effects of LCN2 in disease models associated with systemic inflammation. *Lcn2-KO* animals fed high-fat diets display a consistent increase in food consumption[40], while other reports suggest that LCN2 plays a direct role in insulin sensitivity and glucose metabolism[15,41–43]. Since restricted feeding and caloric restriction are effective in improving metabolic stress during disease[44–46], it is plausible that LCN2 upregulation during diet-induced obesity is an adaptive response that serves to reduce appetite and improve glucose handling. Our data are seemingly consistent with this idea of a prolonged inflammatory state (pancreatic cancer) increasing LCN2 production to regulate appetite. However, our data agree with prior reports suggesting that the reduced caloric intake during pancreatic cancer is detrimental, as the catabolic and energy-wasting programs of cachexia logically require increased caloric intake to achieve organismal energy homeostasis[1,17].

With recent literature demonstrating that LCN2 binds to the MC4R in mediobasal hypothalamic neurons[15], we tested whether disruption of this signaling cascade through ICV injection of AgRP, an inverse agonist of MC4R, would improve appetite during cachexia progression. Similar to *Lcn2-KO* tumor-bearing mice, AgRP-treated mice demonstrated a significant improvement in food intake and modest improvements in muscle mass. Consistent with *Lcn2-KO* tumor-bearing mice, ICV AgRP treatment did not alter ubiquitin-ligase or autophagy-related catabolic pathways in skeletal muscle or cardiac tissue, suggesting that improved caloric intake improves muscle mass through alternative pathways during cachexia. Using an adoptive transfer technique that spares the integrity of the BBB, we demonstrate that reintroducing LCN2 production in the bone marrow compartment alone results in a large induction of LCN2 in the circulation and brain, validating that peripherally derived LCN2 readily crosses the BBB[15]. Furthermore, restoration of *Lcn2* in the bone marrow compartment is sufficient to rescue the cachexia–anorexia phenotype. Since we consistently observe a muscle-sparing effect in tumor-bearing animals devoid of LCN2, we investigated whether this improvement in muscle mass is mediated by improved food intake or through a direct effect of LCN2 on muscle. To address this, we pair-fed tumor-bearing *Lcn2-KO* mice to their tumor-bearing WT counterparts, and observed equivalent skeletal and cardiac tissue mass at the end of the study.

We then explored the biological significance of our observations in human pancreatic cancer. Similar to our preclinical model, we observed a close association between increasing neutrophils and increased circulating LCN2 levels, suggesting that neutrophil expansion is accompanied by increased LCN2 production and secretion. Conversely, the opposite trend was observed in the lymphocyte population, with increased circulating lymphocytes being associated with decreased LCN2 levels. Indeed, increased NLR is associated with worsened morbidity and mortality outcomes for patients with pancreatic cancer[47,48]. For patients that had consecutive positron emission tomographic (with computed tomographic) scans and accompanying blood draws, we performed skeletal muscle and visceral fat analyses at the level of the third lumbar vertebrae to define cross sectional area. Using these data, we identify a close negative correlation between changes in circulating LCN2 levels and loss of both fat and skeletal muscle mass. Since caloric intake is not readily captured in the clinic, we took a logical step in assessing both fat and muscle mass, tissues that are particularly sensitive to prolonged caloric deprivation during chronic disease[49]. Thus, it is conceivable that the fat and lean mass loss we observed in patients with elevated LCN2 levels is partially due to a decrease in appetite and food intake through the molecule's anorectic effects in the brain, as this is the case in our preclinical models. Finally, after dichotomizing pancreatic cancer patients into high (>240 ng/mL) and low (<240 ng/mL) LCN2 levels at diagnosis, we observed that patients with elevated LCN2 at the time of diagnosis had reduced overall survival.

Taken together, key observations we make in our rodent models, including neutrophil expansion, increased circulating LCN2, and lean mass loss, also occur in patients with pancreatic cancer. Indeed, an elevation of the NLR is shown to predict poor prognosis in patients with both resectable and unresectable pancreatic cancer[50,51]. It is hypothesized that an elevation in circulating neutrophils may suppress the activity of cytotoxic T cells and other adaptive immune cell responses, an immunologic shift that could lessen systemic therapeutic efficacy and promote metastatic progression[52]. While most of the literature concerning innate and adaptive immunity in an oncologic setting are focused on immune cells influencing the tumor itself, little is known about how, or if, engagement of the immune system in normal tissue during cancer influences cachexia and tissue wasting. It is clear that an elevation of NLR is associated with poor prognosis in several cancer types, yet, to our knowledge, no mechanistic studies explain (1) how this immunologic shift occurs or (2) what facets of this immunologic shift contribute to,

or improve, disease symptoms. Sarcopenia, or lean mass wasting, is richly described as a negative prognostic marker in nearly all neoplastic diseases, including pancreatic cancer[53]. However, the mechanisms that drive muscle and fat wasting during disease are incompletely understood. Based on our data, we would propose that the increased LCN2 levels, partly as a result of neutrophil expansion and increased expression of *Lcn2* during cachexia, partially explain the increased morbidity and mortality outcomes of cancer patients through its effects on caloric intake and subsequent lean and fat mass wasting.

Here, we have shown that LCN2 is elevated during pancreatic cancer in humans and mice and induces appetite suppression through its actions in the CNS. However, a limitation of this study is that it is unclear if LCN2 regulates appetite during other chronic diseases associated with cachexia, or if there are permissive factors in the appetite-regulating effects of LCN2 that are specific to pancreatic cancer. Although we do not observe a difference in disease burden between WT and *Lcn2-KO* groups through terminal tumor mass or histologic features of the cancer, it is possible that LCN2 affects tumor growth as described previously[12]. The orthotopic model we employed is rapidly progressive, and may not allow enough time to recapitulate this finding. However, since we and others observe an independent effect of LCN2 on appetite when administered to the CNS of healthy mice[15], and if LCN2 blockade also mitigates tumorigenesis, these observations only bolster the notion that targeting LCN2 during pancreatic cancer could prove beneficial through multiple therapeutic means. Although deletion of LCN2 improves cachexia–anorexia, direct pharmacologic inhibition of LCN2 in the CNS during cachexia is outstanding, particularly in the context of the putative LCN2-MC4R axis during disease. While we believe the majority of LCN2 during cachexia is neutrophil-derived, it is conceivable that our BMT experiments also repopulated bone-resident cells[54,55]. In this case, it is plausible that bone-derived LCN2, as described previously[15], contributes significantly to circulating levels during cachexia.

In summary, the data presented here illustrate that pancreatic cancer-associated LCN2 mediates appetite suppression through its actions in the CNS. Despite having recognized cachexia as a serious medical problem for over a thousand years[56], effective therapies remain elusive. The development of cancer ominously influences homeostatic control of energy balance through alterations in resting metabolic tone and, in many cancers associated with cachexia, energy intake. It stands to reason that prolonged anorexia, a sickness behavior frequently observed in patients suffering from cachexia, is not a sustainable strategy for combating neoplastic disease and contributes to ultimate mortality. While improving nutritional intake is likely to improve quality of life and, in some cases, lessen lean and fat wasting, a 2011 international consensus recognized that conventional nutritional support alone is inadequate in treating cachexia[1,2]. Thus, as with most diseases associated with cachexia, we believe addressing nutritional deficiencies is an important facet of treating cachectic patients, but is not a comprehensive approach for improving all cachexia symptoms, including muscle and fat wasting. Nevertheless, identification of factors that influence this seemingly maladaptive anorectic response to pancreatic cancer remains an important component of improving patient outcomes. Based on our data presented here, we propose LCN2 as a potential therapeutic target for the improvement of appetite during pancreatic cancer cachexia.

## Methods

**Mice.** C57BL/6J WT (JAX catalog number 000664), Lipocalin 2 knockout (*Lcn2-KO*, JAX catalog number 024630), Melanocortin 4 receptor knockout (*Mc4r-KO*, JAX catalog number 032518), Interleukin 6 knockout (*Il-6-KO*, JAX catalog

number 002650), and Myeloid differentiation primary response gene 88 knockout (*Myd88-KO*, JAX catalog number 009088) mice were purchased from The Jackson Laboratory (Bar Harbor, ME) and maintained in our animal facility. All mouse strains used in this paper were generated on a C57BL/6J background. All mice were housed and bred in a dedicated mouse room with a temperature 26 °C with a 12-h light/dark cycle and 40% humidity. Animals were provided ad libitum access to food and water (Rodent Diet 5001; Purina Mills) unless otherwise stated. Mice were genotyped according to the standard protocol from The Jackson Laboratory. Sex-, age-, and body-weight-matched WT and *Lcn2-KO* (as well as *Il6-KO* and *Myd88-KO*) mice at 7–10 weeks old were used unless otherwise stated. Only age- and sex-matching was possible with *Mc4r-KO* due to their profound obesity phenotype. In behavioral studies, animals were individually housed for acclimation at least 7 days prior to procedures. Except in survival studies, tumor-bearing animals were euthanized according to the end points of tumor study policy. Mouse studies were conducted in accordance with the National Institutes of Health Guide for the Care and Use of Laboratory animals, and approved by the Institutional Animal Care and Use Committee of Oregon Health & Science University.

**Human samples.** Plasma samples from men and women age 40–82 years old diagnosed with pancreatic cancer were procured through the Brenden Colson Center for Pancreatic Care (Portland, Oregon) through the Oregon Pancreatic Tumor Registry (OPTR). These samples were collected at the time of diagnosis, and for some patients, in follow-up visits. Age- and sex-matched control samples from patients with no clinical evidence of disease were procured from the Oregon Clinical and Translational Research Institute. Specifically, patients that were seen at OHSU with a similar clinical work-up as the pancreatic cancer group, but deemed to have no evidence of pancreatic disease, were included in this study. Informed consent was obtained by these respective organizations for all samples and data utilized in this study. Blood was drawn through venipuncture and plasma and the buffy coat were separated. Samples were then stored at −80 °C until assayed. Given the retrospective and anonymized nature of the human samples and data reported in the manuscript, these studies were deemed nonhumans research, and IRB approval was waived by the OHSU IRB.

**Pancreatic ductal adenocarcinoma cachexia models.** The five separate models of PDAC cachexia utilized herein are clonogenic cell lines derived from separate C57BL/6J mice with pancreatic-specific conditional alleles KRAS$^{G12D}$ and TP53$^{R172H}$ expression driven by the PDX-1-Cre promoter. Since the KPC model is a highly characterized and published model of PDAC cachexia due to its close biological semblance of human disease, we performed all subsequent studies (Figs. 2–6) utilizing this model generously provided by Dr Elizabeth Jaffee[17,21,57,58]. The remaining four PDAC cell lines were generously provided by Drs David Tuveson (FC1242, FC1199, and FC1245) and Robert Vonderheide (4662). All cell lines were maintained in RPMI 1640 supplemented with 10% fetal bovine serum, 1% minimum essential medium non-essential amino acids, 1-mM sodium pyruvate, and 50-U/mL penicillin/streptomycin (Gibco), in cell incubators maintained at 37 °C and 5% $CO_2$. All cell lines were routinely tested and confirmed negative for mycoplasma contamination. Under isoflurane anesthesia, a small surgical incision was made in the upper-left quadrant of the abdomen, reflecting the skin layers, fascia, and muscle wall to expose the pancreas. The tail of the pancreas was injected with either one million cancer cells suspended in 40 μl of PBS or an equal volume of cell-free PBS. After tumor implantation, the abdominal wall and fascia layers were sutured, followed by two surgical skin clips to close the incision site.

**Analysis of cachexia.** Food intake, body mass, and post-procedure health status were monitored daily, with sifting of bedding to collect spilled food. Voluntary wheel running was measured continuously utilizing low-profile running wheels (Med Associates Inc) and lab-constructed wheels. In pair-feeding experiments, *Lcn2-KO* mice were implanted with KPC cells 1 day behind to WT KPC mice, and were pair-fed daily with the amount of food that WT KPC mice consumed within the previous 24 h. Survival in KPC mice was observed twice daily until death, and tumor appearance was confirmed by necropsy. Necropsy tissue analysis included tumor, gastrocnemius, and heart mass by observers blinded to treatment groups. In addition, hypothalamus, heart, gastrocnemius, and liver tissues were immediately flash-frozen for gene expression analysis.

**NMR imaging.** NMR measurements were taken at baseline (day of tumor implantation), as well as early (day 4), mid (day 8), and late cachexia (day 11) according to the manufacturers protocol (EchoMRI LLC, Houston, TX).

**Fecal analyses.** Fecal lipids were harvested and analyzed using a protocol modified from Kraus et al.[59]. Briefly, cage-collected feces were collected using a sieve and pulverized using a pepper grinder. Finely ground feces then underwent a chloroform:methanol (2:1) extraction via centrifugation at $1300 \times g$ for 20 min at RT. After the phase reaction, the organic phase containing the extracted lipids was collected and allowed to evaporate.

Fecal protein concentration was determined using a procotol adapted from Danai et al.[7]. Briefly, 10 mg of pulverized feces was resuspended in 500 μL of lysis buffer (2% SDS, 150-mM NaCl, 0.5-M EDTA), sonicated, and centrifuged at

$13,000 \times g$ for 15 min at 4 °C. The protein concentration was then assessed using a BCA assay according to the manufacturer's instructions.

Fecal protease activity was determined from fresh terminal colonic feces at the time of sacrifice. Briefly, feces were suspended to a 10-mg/mL suspension in Protein Buffer A (0.1% Triton X-100, 0.5-M NaCl, 100-mM $CaCl_2$), homogenized and sonicated, then centrifuged at 14,000 RPM for 15 min. One hundred microliter of the resulting supernatant was added to 200 µL of 3% azocasein, vortexed briefly and incubated for 1 h at 37 °C. After incubation, 500 µL of 8% tri-chloroacetic acid was added to each sample, vortexed, centrifuged at 9000 RPM for 5 min, and the resulting supernatant was assayed for absorbance at 366 nm.

**Bone marrow transplantation (BMT)**. WT and *Lcn2-KO* mice aged 8–12 weeks were administered the alkylating chemotherapeutic treosulfan (Ovastat®, a generous gift provided by Joachim Baumgart at Medac GmbH, Germany) at a dose of 1500 mg/kg/day for 3 days prior to BMT. Donor bone marrow was harvested from 8- to 12-week-old WT, *Lcn2-KO*, or Ly5.1-EGFP mice from femurs, tibias, and humeri by flushing dissected bone cavities with Isocove's modified Dulbecco's medium supplemented with 10% FBS. Bone marrow preparations were then treated with RBC lysis buffer and filtered across a 70-µm cell strainer. $3 \times 10^6$ cells were resuspended in 200-µL HBSS and injected through the tail vein of recipient mice. After initiation of treosulfan treatment, mice received amoxicillin supplemented water (150 mg/L) for 2 weeks to prevent infection. All mice were allowed a minimum of 8 weeks of recovery after BMT to ensure successful bone marrow engraftment with daily monitoring for signs and symptoms of graft rejection. Chimerism was determined at the end of animal experiments by blood flow cytometry analysis.

**ICV cannulation and injections**. Mice were anesthetized using isoflurane and gently placed on a stereotactic alignment instrument (Kopf Instruments). Using sterile technique, bregma was exposed with a 3-mm incision and a 26-gauge lateral ventricle cannula was placed at 1.0-mm X, −0.5-mm Y, and −2.25-mm Z relative to bregma. Cannulas were secured to the skull with embedded screws and cross-linked flash acrylic. Mice were allowed 1 week for recovery after cannulation surgery. Recombinant mouse LCN2 (R&D Systems, 40 ng), AgRP (Phoenix Pharmaceuticals, 1 nmol), and MT-II (Phoenix Pharmaceuticals, 1 nmol) were diluted in artificial CSF and injected in a total volume of 2 µL. For repeated or single injection experiments of LCN2, MT-II, and vehicle, mice received injections under restriction, and received restriction training 5 days prior to initial injections. For ICV AgRP studies, all mice were conditioned to isoflurane anesthesia for 4 days prior to AgRP or vehicle injections.

**CSF extraction**. Mice were anesthetized using isoflurane and placed on a stereotactic alignment instrument (Kopf Instruments). A 2-cm incision was made over the cisterna magna and the trapezius and paraspinal muscles were reflected. Blood and extracellular fluid lying over the cisterna magna were carefully removed to avoid CSF contamination. A glass micropipette (tip diameter of ~400–800 µm) was stereotactically inserted into the cisterna magna for capillary action-based CSF collection.

**Histology and immunohistochemistry**. Mice were deeply anesthetized using a ketamine/xylazine/acepromazine cocktail and sacrificed by transcardial perfusion with 20-mL PBS followed by ice-cold 4% paraformaldehyde (PFA). Tissues were post-fixed in 4% PFA overnight at 4 °C prior to sectioning protocols. Tumor and inguinal white-adipose tissue samples: paraffin-embedded histological sections were stained for hematoxylin and eosin, followed by followed by 10-µM cryostat sectioning. Bone marrow samples: transferred to 70% ethanol for 4 h prior to a gentle decalcification (Decal™, StatLab) for 1 h. Bone was then paraffin-embedded, cryostat sectioned to 50 µM, blocked for 30 min in blocking solution (5% normal donkey serum in 0.01-M PBS and 0.3% Triton X-100), followed by VENTANA staining (Roche) with primary and secondary antibodies (listed below). Brain samples: after post-fixation, brains were cryoprotected in 20% sucrose for 24 h at 4 °C prior to 30-µM microtome sectioning. Free-floating sections were incubated in blocking solution (see bone marrow) for 1 h at room temperature, followed by primary antibody incubation (listed below) overnight at 4 °C. Sections were thoroughly washed with PBS between steps. Sections were mounted on gelatin-coated slides and coverslipped with Prolong Gold anti-fade media (Thermofisher).

Fluorescent-based images were acquired on a Nikon confocal microscope, while chromogen-based images were acquired using a Leica microscope (model DM 4000B). cFos-positive cells were quantified in the PVN of the hypothalamus in three consecutive sections and averaged by a blinded observer.

Primary antibodies utilized above are listed with company, clone, host, species, and concentration defined in parentheses, respectively: LCN2 (R&D Systems, AF1857, goat, 1:800) and cFos (Santa Cruz, sc-166940, goat, 1:25000). A donkey anti-goat AF488 (1:500) secondary antibody from Invitrogen was utilized for fluorescent images, while a horse anti-goat IgG (peroxidase) from VECTOR laboratories (MP-7405-15, 1:800) was used for bone marrow chromogen-based imaging.

**Enzyme-linked immunosorbent assays**. Whole blood was harvested from mice by cardiopuncture, and plasma was isolated using $K_2$EDTA tubes (BD 365974). Mouse plasma and CSF LCN2 concentrations were assayed by ELISA according to the manufacturer's protocol (R&D Systems, Catalog # DY1857). Human plasma LCN2 concentrations were assayed by ELISA according to the manufacturer's protocol (R&D Systems, Catalog # DY1757).

**Flow cytometry**. Mice were anesthetized using a ketamine/xylazine/acepromazine cocktail, and whole blood was collected by cardiopuncture. Two hundred microliter of whole blood was then treated with 1x RBC lysis buffer (Invitrogen) for 10 min then incubated in 100 µL of PBS containing antibodies for 45 min at 4 °C. For intracellular LCN2 staining, cells were then fixed and permeabilized for 10 min, washed with 1x permeabilization buffer (Invitrogen), and stained with primary antibody for 45 min at 4 °C. Cells were stained with a secondary antibody for 1 h at 4 °C and resuspended in 200 µL of RPMI + 5% FBS for analysis. Cells were thoroughly washed between steps with either RPMI + 5% FBS (prior to fixation/permeabilization) or 1x permeabilization buffer (after fixation/permeabilization).

Cells were gated on LD, SSC, and FSC singlet. Immune cells were defined as CD45 + cells. Myeloid cells were defined as CD45highCD11b+, and lymphocytes were defined as CD45highCD11b−. Myeloid cells were further gated into Ly6Clow monocytes (Ly6ClowLy6G−), Ly6C$^{high}$ monocytes (Ly6C$^{high}$Ly6G−), and neutrophils (Ly6C$^{mid}$Ly6G+). Lymphocytes were grouped as either CD3+ T cells or CD19+ B cells. The resulting T-cell gate was further stratified into CD4+ or CD8+ T cells. All of the aforementioned groups were intracellularly stained for LCN2. Flow cytometry analysis was performed on the LSRII analytic flow cytometer.

Antibodies and reagents utilized for flow cytometry analysis can be found in Supplementary Table 4. All flow cytometry was performed on a BD LSR II, and data were analyzed in FlowJo software.

**Quantitative real-time PCR**. Snap-frozen tissues were rapidly homogenized, and RNA was purified with the RNeasy Mini Kit (Qiagen). Samples were then reverse-transcribed with the High Capacity cDNA Reverse Transcription Kit (Life Technologies). qRT-PCR was performed using reagents, and TaqMan primer probes listed in Supplementary Table 5. Tissues were normalized to 18S using the ddCT method.

**Western blot**. Protein was extracted from snap-frozen tissues by homogenization followed by brief sonication. Sixty microgram of protein was loaded in each lane and run on Novex 6–18% Tris-Glycine gels (Life Technologies). Gels were transferred to PDVF membranes (Millipore) and blocked with 5% BSA for 1 h. Membranes were incubated with primary antibodies overnight at 4 °C with gentle agitation. Blots were then washed with TBST and incubated in secondary antibodies for 1 h prior to imaging using the Li-Cor Odyssey imaging system. A complete list of antibodies can be found in Supplementary Table 6.

**RNA-seq of circulating neutrophils**. Whole blood was isolated from mice by cardiopuncture, and total RNA was isolated from FACS-sorted CD11b$^{high}$Ly6G$^{high}$ neutrophils using an RNAeasy Plus Micro kit (Qiagen). Integrity of RNA was verified by an Agilent Bioanalyzer prior to cDNA preparation using the SMART-Seq v4 Ultra Low Input kit (Takara). RNA libraries were prepared using a TruSeq DNA Nano kit (Illumina) and verified by Tapestation (Agilent). Library concentrations were measured by real-time PCR with StepOnePlus Real Time PCR System (ThermoFisher) followed by a Kapa Library Quantification Kit (KapaBiosystems/Roche). Libraries were then sequenced with a 100-cycle single-read protocol using a HiSeq 2500 sequencer (Illumina). Quality control checks were performed using the FastQC package, and raw reads were normalized using the DESeq2 Bioconductor package.

**Computed-tomography-based body composition analysis**. Access to CT scans was approved by the OHSU IRB, as all patients were consented to the OPTR protocol. For each patient, a single contrast-enhanced axial image at the third lumbar vertebrae (L3) was selected, anonymized, and saved in DICOM format using Osirix v11.0 (Pixmeo). The images underwent first-pass segmentation by an automated algorithm in MATLAB R2016a (MathWorks)[60], which labels skeletal muscle and visceral fat based upon its pre-established radiodensity range (−29 to +150 and −190 to −30 Hounsfield Units, respectively)[24,61]. The images were then processed with Sliceomatic 5.0 (Tomovision) for manual corrections. Skeletal muscle index (SMI) and normalized visceral fat measures for each patient were calculated by dividing total skeletal muscle and fat cross-sectional area at L3 (cm²) by patient height squared (m²).

**Statistics**. All statistical analyses for murine data were performed in GraphPad Prism 8.0 software. Quantitative data are reported as mean ± standard error. Two-tailed Students *t* tests were performed when comparing two groups. When comparing more than two groups of a single genotype, One-way ANOVA was utilized. Correlation analysis were performed after assessment of normality using Shapiro–Wilk tests, demonstrating that all data analyzed followed a Gaussian distribution (Pearson correlation, parametric data). Two-way ANOVA with Bonferroni multiple comparisons test was utilized when comparing multiple genotypes

and treatment groups (sham and tumor) unless otherwise specified in figure legends. Data from human studies from exported excel files were analyzed using both GraphPad Prism 8.0 and IBM SPSS Statistics Suite (version 25), with Kaplan–Meier survival curve comparison using log-rank Mantel–Cox test. Determination of dichotomizing cutpoints of plasma LCN2 levels and survival outcomes utilized the Evaluate Cutpoints adaptive algorithm software in RStudio as described previously[25]. For all analyses, a $p$ value of <0.05 was considered to be statistically significant. For histochemistry analyses, images were representative of at least three separate stainings. Western blot images are representative of at least two separate experiments. All measurements were from distinct samples and not taken from the same sample more than once. When applicable, all statistical tests were performed as two-tailed analyses.

**Reporting summary**. Further information on experimental design is available in the Nature Research Reporting Summary linked to this paper.

## Data availability

All data associated with this study are available in the main text, supplementary materials, or source data file. There are no restrictions on data availability. FCS flow cytometry files are available upon reasonable request. Sequencing data are available at GEO (GSE150061). Source data are provided with this paper.

## Code availability

Custom code associated with this manuscript, created by Ogulszka and colleagues, is publicly available through their associated manuscript[25].

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

## Acknowledgements

We thank members of the Oregon Clinical and Translational Research Institute and Brenden Colson Center for Pancreatic Care for providing pancreatic cancer blood samples. We thank Drs Elizabeth Jaffee (KPC), David Tuveson (FC1242, FC1199, and FC1245), and Robert Vonderheide (4662) for graciously providing the syngeneic pancreatic cancer cell lines used for our studies. The graphical abstract and Fig. 4a were made using BioRender (BioRender.com). Finally, we thank Ashley J Olson, PA-C for technical assistance with the ICV AgRP study. This work was supported by NIH R01CA217989 (Marks), the Brenden-Colson Center for Pancreatic Care at OHSU (Marks), NIH K08CA245188 (Grossberg), and NIH F30CA254033 (Olson).

## Author contributions

B.O. and D.L.M. designed the study, analyzed the data, and wrote the manuscript with input from the other authors. B.O., X.Z., M.A.N., P.R.L., J.T.B., A.B., K.R.P., H.M., and S. M.K. performed experiments and analyzed data. K.G.B., K.A.M., J.E., and A.J.G. contributed with discussion, data abstraction, and data interpretation.

## Competing interests

D.L.M. is a consultant for Pfizer, Inc. and Alkermes, Inc. D.L.M. is a consultant, has received grant funding, and has equity in Tensive Controls, Inc. The other authors declare no competing interests.
