## [Peer Review File · Nature Communications]

REVIEWER COMMENTS

Reviewer #1 (Remarks to the Author):

The manuscript by Olson and co-authors examines the hypothesis that neutrophil derived Lipocalin 2 (LCN2) drives pancreatic cancer associated anorexia via MC4R signaling. The authors present data that demonstrate elevated circulating LCN2 in murine models of and patients with pancreatic cancer. In mice, they trace LCN2 production to the neutrophil population. They demonstrate that elevated LCN2 is associated with reduced food intake and muscle mass during cancer progression, when cancer associated cachexia ensues, the latter supported by clinical association of reduced skeletal muscle index in patients with progression of cancer and elevation of LCN2. By intracerebroventricular LCN2 injection in mice, they induce increased neuronal activity, presumably in an MC4R-dependent manner, which is associated with reduced food intake. Normalization of food intake occurs on cessation of LCN2 injection. When cachexia inducing tumors are implanted in Lcn2-KO mice, the authors observe improved activity, lessened anorexia, but no preservation of muscle mass.

In summary, the authors show that LCN2 elevations contributes to the anorexia that is associated with cancer-cachexia. As the authors acknowledge, the finding that LCN2 reduces appetite is not new, but it has not been examined in the context of cancer progression.

The authors may wish to consider the following points to strengthen their work:

- 1) What is the factor from pancreatic cancer that causes neutrophils to upregulate LCN2? An answer to this question would add a great deal of interest to the work.
- 2) I have trouble understanding why they focus on neutrophil LCN2, as it is highly expressed to the same level in sham animals as in tumor bearing animals. If there is no change between sham and cachectic mice, how can it be the driving force behind their phenotypes? Could they knock out LCN2 only in neutrophils to see if that prevents the anorexia phenotype.
- 3) The authors should verify that the increase in PVN cFos expression is unique to MC4R+ neurons by double IHC. Right now, it just seems like there is a general increase in PVN cFos. The PVN contains many neuromodulator populations (oxytocin, vasopressin, corticotropin releasing factor, GLP1, and others) that could be affected to influence the observed phenotype.
- 4) In line with point 2, if the authors think that MC4R+ neurons in the PVN are critical for the phenotype they observe, they should see if LCN2 infusion (ICV) has any effect on food intake in MC4R-KO mice or those with PVN MC4R neurons ablated (e.g., cross Jax # 030759 with iDTR mouse jax# 007900).
- 5) Data presentation and analysis:
 - 5.1) Figure 1 K-N: The pooled analysis across murine models is potentially problematic, because of the risk of batch effects of models (one model may have generally high LCN2 and low food intake, whereas another may have low LCN2 and high food intake). The authors should analyze the data for each model system (and present those findings separately) and then meta-analyze across the model systems. Some model systems seem to show increased food intake with increasing LCN2 levels.
 - 5.2) In Figure 3, it is unclear that a 2-way ANOVA was done, as all 4 groups are not compared graphically. I suggest using an A, B, AB, etc... labeling system to delineate significant post-hoc findings.

5.3) They may wish to consider adding legends/more info to their figures, especially the gene expression data as it is hard to tell what organ/tissue gene expression is being shown without looking deep into the figure legend.

Reviewer #2 (Remarks to the Author):

Summary

The current study by Olson et al. explores the role of Lcn2, a circulating mediator of inflammatory and metabolic pathways, in pancreatic cancer cachexia. In this respect, the authors demonstrate that circulating levels of Lcn2 were elevated in different sc transplantation mouse models of pancreatic cancer-induced cachexia, correlating with decreased food intake and reduced skeletal muscle mass. In cachexia, Lcn2 production was strongly elevated in lung, liver and spleen as well as in circulating neutrophils. Tumor-bearing Lcn2 KO mice displayed elevated food consumption and spared cardiac muscle wasting as compared to WT counterparts, while immunological profiles, skeletal muscle and liver gene expression remained unaltered. Based on previous literature, demonstrating a ligand function of Lcn2 for the hypothalamic MC4R, ICV infusion of the MC4R inverse agonist AgRP led to an improvement of food intake and skeletal as well as cardiac muscle mass in cachectic animals. Upon bone marrow transplantation into WT or Lcn2 KO mice, restoration of bone marrow Lcn2 rescued circulating and central Lcn2 levels, demonstrating that bone marrow-derived Lcn2 indeed crossed the blood-brain-barrier. Rescue of Lcn2 levels by WT bone marrow transplantation into Lcn2 KO mice led to a suppression of cumulative food intake as compared to the KO/KO situation. Finally, Lcn2 levels were elevated post-diagnosis in pancreatic cancer patients as compared with healthy controls, correlating with neutrophil numbers and skeletal muscle loss. The authors conclude that Lcn2 is significantly elevated in both murine and human pancreatic cancer and mediates appetite suppression and loss of lean muscle mass through a melanocortin-dependent mechanism.

General comments

Cancer cachexia represents an unmet clinical need. Due to its complexity, insights into novel pathomechanisms and their potential therapeutic applications are direly needed to improve the patient's clinical situation. In this respect, the current manuscript by Olson et al. addresses an important and interesting topic in biomedical research. Overall, the manuscript is concise, well written and structured, making a convincing case for the elevation of Lcn2 levels in both murine and human pancreatic cancer. Another strength resides in the use of state-of-the art technologies to approach the author's scientific questions.

However, a number of major concerns exist that require additional attention by the authors: a) In a number of cases, the claimed effects on cachectic phenotypes, i.e. skeletal muscle mass, are minimal or simply non-existing, e.g. there is no difference in skeletal muscle mass upon Lcn2 KO in Fig. 3C, no effect on survival in Fig. S2D, no effect on skeletal muscle mass upon Lcn2 KO in Fig. S2J, no impact of WT bone marrow transplantation into Lcn2 KO mice on total food intake (Fig. 5E) and skeletal muscle mass in Fig. 5F, and as a consequence of the absence of any robust effects of Lcn2 KO on skeletal muscle mass, also the interpretation of the pair-feeding study in Fig. 6 becomes difficult. Overall, the authors develop their story on the basis of either minimal or non-significant effects that overall question the biological relevance of the observed effects.

b) The phenotypic characterization of the animal studies is premature. It is particularly surprising that the manuscript exclusively focuses on skeletal/cardiac muscle as an indicator for cachexia. In fact, many studies have shown that adipose tissue loss is also a key component of this disease, and particular changes in food consumption are expected to be reflected by alterations in body weight and body fat content. Indeed, the study by Mosialou et al. (Nature 2017) clearly demonstrated effects on body fat upon Lcn2-MCR4 activation but observed no effects on lean muscle mass. Consequently, the authors need to provide a more detailed phenotypic characterization of their animal models by providing data on body weight, body fat content, fat depot size and histology, energy expenditure, and fecal energy excretion to have a complete view on energy homeostasis. Also, the same study implicated a direct impact of circulating Lcn2 on pancreatic islet function. Thus, the current manuscript would substantially benefit from some basic measurements of insulin secretion etc., particularly given that also cancer cachexia represents a condition of insulin resistance. A more complete phenotypic characterization would also answer the question whether or not Lcn2 may impose tissue-selective effects in the context of cancer cachexia.

c) Data on the central functions of Lcn2 in appetite control via MCR4 have been recently published (Mosialou, Nature 2017), thereby limiting the novelty aspects of Fig. 3, 4, S4. In this respect, the author's conclusion and statement that Lcn2 acts via MCR4 in cachectic appetite regulation is not backed up by experimental data as only AgRP was used to counteract tumor-induced cachexia in Fig. 4, no links were made to Lcn2 signaling. Lcn2 ICV administration should be able to worsen cachectic phenotypes in MCR4 WT animals but not do so in MCR4 KO mice. These or similar rescue experiments have to be provided to make a clear case for an Lcn2-MCR4 axis in cancer cachexia.

Minor comments

1. The cited Nature 2017 paper concludes that bone marrow is not a major source of Lcn2. Please discuss the discrepancy.
2. Fig. 3I-L: Please show data from all 4 experimental groups. Otherwise, it is difficult to judge on the degree of gene induction upon tumor implantation.
3. Please provide more experimental details on the sc tumor cell transplantations (no of cells etc).
4. Fig. S4: Please show data on skeletal muscle and adipose tissue mass.
5. How do you explain any (even if minimal) effect on muscle mass without alterations in proteasomal markers etc?

Reviewer #3 (Remarks to the Author):

The authors have used a combination of in-vivo murine and human studies to identify LCN2 as a potential mediator of reduced food intake and subsequent reduced lean mass in cachexia associated with pancreatic cancer. The investigation of appetite suppression in cachexia is a neglected area, and the findings of the present study are novel and of potential clinical and scientific importance. The methodological and statistical techniques used are robust. However, I do have some major and minor comments.

Major Comments

1. The authors initially analyse 5 different PDAC murine models, but then concentrate further studies purely in the KPC model. Their initial results confirm moderate heterogeneity in the nutritional and biochemical characteristics of the different PDAC models. Although the authors give a short justification of their subsequent sole usage of the KPC model in the Methods, this methodology also requires greater emphasis in the Results and Discussion to

make it clear to the reader that the observed findings are confined to one model with the associated limitations.

2. The authors state that the bone marrow is the predominant site of LCN2 “production” (lines 163 onwards), but I would argue that only LCN2 expression (rather than production) has been assessed. The exact site and timing of LCN2 production during the neutrophil’s short half-life appears still unclear, especially as the spleen, liver and bone marrow are all involved in neutrophil clearance, as well as haematopoiesis. Equally, bone marrow LCN2 expression was similar between cachexia and sham. Suggest change “production” to “expression”.

3. The mechanistic MC4R data do not specifically involve the use of LCN2 as a ligand and therefore, although good to note (as LCN2 is reported to be a ligand of MC4R), they do not add significantly to the overall manuscript. The senior author (and other groups) have previously published on the role of MC4R antagonism/inverse agonism as potential therapeutic methods of cachexia amelioration. Do the authors have any novel data on the central administration of LCN2 to mice or the interaction between LCN2 and MCR4?

4. The section on human studies is important but is currently limited in scope and requires expansion. Further patient details are required, including demographics, age, stage etc. How many follow-up assessments were performed, and what treatments were patients receiving (treatment will influence rate and nature of muscle wasting)? Regarding the CT data, what phase of scans were used (they should be consistent; usually portal phase), and why was delta SMI correlated with delta LCN2 (rather than total levels)? Did the authors analyse other CT parameters of wasting (e.g. measures of subcutaneous and visceral fat)?

5. The authors highlight the expansion of the neutrophil population and decrease of the lymphocyte population in cachexia. However, discussion of the prognostic importance of the NLR (neutrophil-lymphocyte ratio) in cancer is limited. What was the NLR of the recruited pancreatic cancer patients? Or other pro-inflammatory measures (e.g. CRP/modified GPS)? And how do they relate to LCN2 levels?

Minor Comments

Line 108: “muscle-sparing effects of LCN2 blockade is” should be changed to “muscle-sparing effects of LCN2 blockade are”

Line 404: “muscle-sparing effects of LCN2 is” should be changed to “muscle-sparing effects of LCN2 are”

Line 144: “Furthermore, we observe” should be changed to past tense consistent with rest of manuscript

REVIEWER COMMENTS

Reviewer #1 (Remarks to the Author):

The manuscript by Olson and co-authors examines the hypothesis that neutrophil derived Lipocalin 2 (LCN2) drives pancreatic cancer associated anorexia via MC4R signaling. The authors present data that demonstrate elevated circulating LCN2 in murine models of and patients with pancreatic cancer. In mice, they trace LCN2 production to the neutrophil population. They demonstrate that elevated LCN2 is associated with reduced food intake and muscle mass during cancer progression, when cancer associated cachexia ensues, the latter supported by clinical association of reduced skeletal muscle index in patients with progression of cancer and elevation of LCN2. By intracerebroventricular LCN2 injection in mice, they induce increased neuronal activity, presumably in an MC4R-dependent manner, which is associated with reduced food intake. Normalization of food intake occurs on cessation of LCN2 injection. When cachexia inducing tumors are implanted in Lcn2-KO mice, the authors observe improved activity, lessened anorexia, but no preservation of muscle mass.

In summary, the authors show that LCN2 elevations contributes to the anorexia that is associated with cancer-cachexia. As the authors acknowledge, the finding that LCN2 reduces appetite is not new, but it has not been examined in the context of cancer progression.

The authors may wish to consider the following points to strengthen their work:

1) What is the factor from pancreatic cancer that causes neutrophils to upregulate LCN2? An answer to this question would add a great deal of interest to the work.

This is an excellent question, and we believe there are several inflammatory mediators secreted not only directly from the pancreatic tumor, but also from distant organs influenced by the progression of disease, that ultimately mediate the large induction of LCN2 in this model. The induction of LCN2 has been attributed to several canonical inflammatory cytokines, including IFN γ ¹, TNF α ¹, IL-1 β ², and IL-6³ to name a few. Thus, we believe LCN2 is likely an inflammation-induced mediator of sickness during cachexia. Since both IL-6⁴ and MyD88⁵ (the universal adaptor protein to all Toll-like receptors [except TLR3] and the interleukin 1 receptor family) signaling are inflammatory pathways known to play roles in cancer cachexia, we performed new experiments in the revised manuscript that examines circulating LCN2 levels in both IL-6 and MyD88 knockout mice. We observe significant decreases in circulating LCN2 both IL-6KO and MyD88KO tumor-bearing mice when compared to WT tumor-bearing mice (Supplemental Figure 2B-C). While both IL-6 and MyD88 signaling induce LCN2 in our model of pancreatic cancer cachexia, there are likely several other inflammatory mediators that mediate the large induction of LCN2 in this model that extend beyond the scope of this article.

2) I have trouble understanding why they focus on neutrophil LCN2, as it is highly expressed to the same level in sham animals as in tumor bearing animals. If there is no change between sham and cachectic mice, how can it be the driving force behind their phenotypes? Could they knock out LCN2 only in neutrophils to see if that prevents the anorexia phenotype.

This point is now further elucidated in the discussion and results section of the manuscript (lines 581-590 of unmarked manuscript). We conclude that large systemic production of LCN2 is predominantly neutrophil derived through the following lines of evidence: 1) LCN2, or Neutrophil Gelatinase Associated Lipocalin (NGAL), was historically described as a neutrophil-derived protein that is stored in secretory granules and secreted once stimulated^{6,7}; 2) although we detect a modest, but not significant elevation of intracellular LCN2 in tumor-bearing mice (Figure 2F, left bar graph figure), the large increase in circulating neutrophils combined with this modest intracellular elevation creates a massive elevation of relative intracellular abundance of LCN2 in tumor-bearing mice (Figure 2F, right fluorescent intensity histogram of LCN2 staining intensity); and 3) the significant upregulation of LCN2 transcripts in neutrophils from tumor-bearing mice compared to sham mice as detected by neutrophil RNA-sequencing (Figure 2G). These observations are consistent with neutrophils being a major contributor to the overall elevation of systemic LCN2 during pancreatic cancer cachexia. However, we also discovered that other immune cells increase their protein production of LCN2 during pancreatic cancer cachexia (Supplemental Figure 2E), which likely contributes to the systemic elevation of LCN2 during disease. When comparing total intracellular LCN2 levels across myeloid and lymphoid lineages (including non-neutrophil myeloid, T-cells, and B-cells), the total intracellular fluorescent intensity when probing for LCN2 is magnitudes higher in neutrophils than the other immune cell populations (Supplemental Figure 2E). Finally, while analysis of equal numbers of neutrophils between sham and PDAC groups demonstrates only a modest increase in the cancer group, the massive release of neutrophils from the marrow compartment in this model does not support the conclusion that this molecule is highly produced at the same level in sham animals as in tumor groups.

Although it would be possible to knock out LCN2 from granulocytes by crossing LCN2^{loxP} (Jax # 031034) with the MRP8-Cre-ires/GFP mouse (Jax # 021614), the LCN2 floxed animal is cryopreserved. Thus, establishing this model and breeding enough mice to perform a powered study is likely to take well over a year. Finally, while identifying the precise cellular mediators of LCN2 production during pancreatic cancer would be interesting, we do not think this knowledge would change a potential therapeutic approach in targeting LCN2 for improving appetite.

3) The authors should verify that the increase in PVN cFos expression is unique to MC4R+ neurons by double IHC. Right now, it just seems like there is a general increase in PVN cFos. The PVN contains many neuromodulator populations (oxytocin, vasopressin, corticotropin releasing factor, GLP1, and others) that could be affected to influence the observed phenotype.

This is an important point to address, and while this experiment describing PVN cFos expression in MC4R neurons is reported in full by Mosialou et al. (Nature 2017) in Figure 5F of their manuscript, we go beyond this and now address the behavioral component of this question in our revised manuscript. Specifically, we repeatedly administer LCN2 to the brains of WT and MC4RKO mice and observe sustained significant decrease in food consumption, body mass, fat mass, and lean mass of WT mice compared to MC4RKO mice (Supplemental Figure 5H-L).

4) In line with point 2, if the authors think that MC4R+ neurons in the PVN are critical for the phenotype they observe, they should see if LCN2 infusion (ICV) has any effect on food intake in MC4R-

KO mice or those with PVN MC4R neurons ablated (e.g., cross Jax # 030759 with iDTR mouse jax# 007900).

Please see our comments to #3, as we believe this experiment readily addresses this concern.

5) Data presentation and analysis:

5.1) Figure 1 K-N: The pooled analysis across murine models is potentially problematic, because of the risk of batch effects of models (one model may have generally high LCN2 and low food intake, whereas another may have low LCN2 and high food intake). The authors should analyze the data for each model system (and present those findings separately) and then meta-analyze across the model systems. Some model systems seem to show increased food intake with increasing LCN2 levels.

We now present linear regression analysis for each individual Figure 1K-N in Supplemental Figure 1. With only 4-6 mice per group, providing meaningful correlation data is difficult with this level of experimental power. However, all models but FC1199 and FC1245 (plasma levels only) display trends of decreasing food consumption with increasing LCN2 levels.

While we now report these individual pancreatic cancer models separately in the supplement, we believe including the grouped analysis remains valid. Since the cell lines used are clonogenic and derived from the KRAS^{G12D}TP53^{R172H} x PDX-1-Cre genetically engineered mouse model (KPC-GEMM)⁸, orthotopic implantation of these cell lines result in varying degrees of cachexia despite having a similar biological origin. Therefore, we believe cross-comparisons amongst these models are useful and akin to human disease, in which patients with similarly staged tumors, treatments, and demographics can experience significantly different cachexia symptoms.

5.2) In Figure 3, it is unclear that a 2-way ANOVA was done, as all 4 groups are not compared graphically. I suggest using an A, B, AB, etc... labeling system to delineate significant post-hoc findings.

We now add comparison bars in Figure 3 to illustrate key findings upon 2-way ANOVA that were not otherwise explicit, including the main column effects between tumor-bearing groups in panel A (a key finding of this manuscript).

5.3) They may wish to consider adding legends/more info to their figures, especially the gene expression data as it is hard to tell what organ/tissue gene expression is being shown without looking deep into the figure legend.

A title is now added to all graphs with gene expression data to quickly display which organ or tissue the data is derived from.

Reviewer #2 (Remarks to the Author):

Summary

The current study by Olson et al. explores the role of Lcn2, a circulating mediator of inflammatory and metabolic pathways, in pancreatic cancer cachexia. In this respect, the authors demonstrate that circulating levels of Lcn2 were elevated in different sc transplantation mouse models of pancreatic cancer-induced cachexia, correlating with decreased food intake and reduced skeletal muscle mass. In cachexia, Lcn2 production was strongly elevated in lung, liver and spleen as well as in circulating neutrophils. Tumor-bearing Lcn2 KO mice displayed elevated food consumption and spared cardiac muscle wasting as compared to WT counterparts, while immunological profiles, skeletal muscle and liver gene expression remained unaltered. Based on previous literature, demonstrating a ligand function of Lcn2 for the hypothalamic MC4R, ICV infusion of the MC4R inverse agonist AgRP led to an improvement of food intake and skeletal as well as cardiac muscle mass in cachectic animals. Upon bone marrow transplantation into WT or Lcn2 KO mice, restoration of bone marrow Lcn2 rescued circulating and central Lcn2 levels, demonstrating that bone marrow-derived Lcn2 indeed crossed the blood-brain-barrier. Rescue of Lcn2 levels by WT bone marrow transplantation into Lcn2 KO mice led to a suppression of cumulative food intake as compared to the KO/KO situation. Finally, Lcn2 levels were elevated post-diagnosis in pancreatic cancer patients as compared with healthy controls, correlating with neutrophil numbers and skeletal muscle loss. The authors conclude that Lcn2 is significantly elevated in both murine and human pancreatic cancer and mediates appetite suppression and loss of lean muscle mass through a melanocortin-dependent mechanism.

General comments

Cancer cachexia represents an unmet clinical need. Due to its complexity, insights into novel pathomechanisms and their potential therapeutic applications are direly needed to improve the patient's clinical situation. In this respect, the current manuscript by Olson et al. addresses an important and interesting topic in biomedical research. Overall, the manuscript is concise, well written and structured, making a convincing case for the elevation of Lcn2 levels in both murine and human pancreatic cancer. Another strength resides in the use of state-of-the art technologies to approach the author's scientific questions.

However, a number of major concerns exist that require additional attention by the authors:

a) In a number of cases, the claimed effects on cachectic phenotypes, i.e. skeletal muscle mass, are minimal or simply non-existing, e.g. there is no difference in skeletal muscle mass upon Lcn2 KO in Fig. 3C, no effect on survival in Fig. S2D, no effect on skeletal muscle mass upon Lcn2 KO in Fig. S2J, no impact of WT bone marrow transplantation into Lcn2 KO mice on total food intake (Fig. 5E) and skeletal muscle mass in Fig. 5F, and as a consequence of the absence of any robust effects of Lcn2 KO on skeletal muscle mass, also the interpretation of the pair-feeding study in Fig. 6 becomes difficult. Overall, the authors develop their story on the basis of either minimal or non-significant effects that overall question the biological relevance of the observed effects.

We appreciate this critical assessment of our work, but disagree with the notion that this story is built upon minimal or non-significant effects. As described in the title, "Lipocalin 2 mediates appetite suppression during pancreatic cancer cachexia", we believe the principal effect of LCN2 during pancreatic cancer cachexia is its ability to regulate food consumption, with this increased food

consumption also modestly improving skeletal muscle mass. Robust and significant pieces of data suggesting LCN2 regulates food consumption during pancreatic cancer cachexia can be found in Figures 1K, 1N, 3A, 3J, 3K, S3I, and 5D. We believe the consistent trend in increased muscle mass in *Lcn2*-KO mice stems from this increased food consumption, which is addressed through our pair-feeding study in which *Lcn2*-KO tumor-bearing mice display equal (and even slightly decreased in the case of the heart) muscle mass.

We have also added commentary to the discussion to further clarify the role of nutrition in cancer cachexia: “While improving nutritional intake is likely to improve quality of life and, in some cases, lessen lean and fat wasting, a 2011 international consensus recognized that conventional nutritional support alone is inadequate in treating cachexia^{9,10}. Thus, as with most diseases associated with cachexia, we believe addressing nutritional deficiencies is an important facet of treating cachectic patients, but is not a comprehensive approach for improving all cachexia symptoms, including muscle wasting” (Lines 708-715 of unmarked manuscript).

b) The phenotypic characterization of the animal studies is premature. It is particularly surprising that the manuscript exclusively focuses on skeletal/cardiac muscle as an indicator for cachexia. In fact, many studies have shown that adipose tissue loss is also a key component of this disease, and particular changes in food consumption are expected to be reflected by alterations in body weight and body fat content. Indeed, the study by Mosialou et al. (Nature 2017) clearly demonstrated effects on body fat upon *Lcn2*-MCR4 activation but observed no effects on lean muscle mass. Consequently, the authors need to provide a more detailed phenotypic characterization of their animal models by providing data on body weight, body fat content, fat depot size and histology, energy expenditure, and fecal energy excretion to have a complete view on energy homeostasis. Also, the same study implicated a direct impact of circulating *Lcn2* on pancreatic islet function. Thus, the current manuscript would substantially benefit from some basic measurements of insulin secretion etc., particularly given that also cancer cachexia represents a condition of insulin resistance. A more complete phenotypic characterization would also answer the question whether or not *Lcn2* may impose tissue-selective effects in the context of cancer cachexia.

This is an excellent point, and we believe our experiments focused on further characterizing this model greatly strengthen the revised manuscript. We now provide accompanying data characterizing changes in body mass (Supplemental Figure 3F), body fat content (Figure 3F-IH), inguinal fat pad mass (Figure 3I), fat histology (Supplemental Figure 3E), fat browning gene expression signatures (Supplemental Figure 3A-D), fecal energy excretion (Supplemental Figure 3Q-U), and glucose monitoring (Supplemental Figure 3V-X) in our pancreatic cancer cachexia model.

Briefly, we observed sparing of fat tissue in LCN2KO mice compared to WT mice that was not accompanied by changes in browning genes. Consistent with previous reports¹¹, body mass was not significantly altered in this model of cachexia, likely owing to abdominal ascites and third-spacing edema as the primary tumor develops. We observed no significant alterations in fecal energy excretion as indicated by fecal lipids, proteins, and protease activity. Finally, while we observed a significant increase in glucose levels in LCN2KO tumor-bearing mice, this observation may be due to their improved caloric intake.

c) Data on the central functions of *Lcn2* in appetite control via MCR4 have been recently published (Mosialou, Nature 2017), thereby limiting the novelty aspects of Fig. 3, 4, S4. In this respect, the author’s conclusion and statement that *Lcn2* acts via MCR4 in cachectic appetite regulation is not

backed up by experimental data as only AgRP was used to counteract tumor-induced cachexia in Fig. 4, no links were made to Lcn2 signaling. Lcn2 ICV administration should be able to worsen cachectic phenotypes in MCR4 WT animals but not do so in MCR4 KO mice. These or similar rescue experiments have to be provided to make a clear case for an Lcn2-MCR4 axis in cancer cachexia.

As discussed in a previous review comment, the revised manuscript includes a chronic ICV LCN2 study in both MC4R WT and MC4R KO mice (Supplemental Figure 5H-L). Indeed, we observed a sustained reduction in food intake, body mass, lean mass, and fat mass in MC4R WT mice, but not MC4R depleted mice.

Minor comments

1. The cited Nature 2017 paper concludes that bone marrow is not a major source of Lcn2. Please discuss the discrepancy.

While Mosialou et al. suggest bone marrow is not a major source of LCN2, the paper never measures LCN2 production in the bone marrow compartment. Nonetheless, our limitations section details how our bone marrow transplantation experiments may repopulate bone-resident cells. Briefly, while the bone may be an important mediator of LCN2 levels during physiologic conditions as described by Mosialou et al, the immunologic shift incurred by pancreatic cancer, combined with our immune cell expression data and bone marrow transplantation experiments, suggest this compartment significantly contributes to circulating LCN2 levels during pancreatic cancer cachexia (Figure 5B).

2. Fig. 3I-L: Please show data from all 4 experimental groups. Otherwise, it is difficult to judge on the degree of gene induction upon tumor implantation.

These gene expression data are represented as fold change of tumor-bearing mice over genotype-matched controls. To add clarity to this point, we include the following in the legend: "All gene expression data represented as fold change over genotype-matched controls. No difference was observed in baseline expression of transcripts between WT and *Lcn2-KO* mice."

3. Please provide more experimental details on the sc tumor cell transplantations (no of cells etc).

We now provide more experimental details of the orthotopic tumor implantations, including number of cells, volume of inoculum, and surgical procedure.

4. Fig. S4: Please show data on skeletal muscle and adipose tissue mass.

In this study, since ICV injections were halted at day 14 and animals were monitored for a week thereafter to observe changes in food consumption behavior, the utility of collecting skeletal and fat mass was minimal. Thus, by the study's design, skeletal and adipose tissue mass cannot be meaningfully assessed, as the outcomes were focused on behavioral effects of LCN2 in the brain. However, we now provide the requested lean and fat mass data in the revision experiment in response to comment C (central administration of LCN2 to both WT and MC4RKO mice; Supplemental Figure 5H-L).

5. How do you explain any (even if minimal) effect on muscle mass without alterations in proteasomal markers etc?

This is an interesting point and consistent with the consensus in the cachexia field in that increasing food consumption alone will not completely reverse wasting. In the current study, this is exemplified by our ICV AgRP study (Figure 4), in which stimulation of food intake of tumor-bearing mice improve muscle mass without influencing ubiquitin-ligase and autophagy-related pathways (Supplemental Figure 4). Therefore, it is hypothesized that caloric intake-independent pathways mediate muscle catabolism during cachexia, with literature attributing activation of various muscle catabolic pathways to tumor-derived factors such as HSP70 and HSP70¹², Activin A and IL-6¹³, and TWEAK signaling¹⁴ to name a few. Thus, since we observe no change in tumor or other inflammatory signaling markers in LCN2KO mice, it is likely that these tumor-muscle crosstalk pathways (amongst other signaling axes) are significant mediators of muscle catabolism in this model of cancer cachexia.

Reviewer #3 (Remarks to the Author):

The authors have used a combination of in-vivo murine and human studies to identify LCN2 as a potential mediator of reduced food intake and subsequent reduced lean mass in cachexia associated with pancreatic cancer. The investigation of appetite suppression in cachexia is a neglected area, and the findings of the present study are novel and of potential clinical and scientific importance. The methodological and statistical techniques used are robust. However, I do have some major and minor comments.

Major Comments

1. The authors initially analyse 5 different PDAC murine models, but then concentrate further studies purely in the KPC model. Their initial results confirm moderate heterogeneity in the nutritional and biochemical characteristics of the different PDAC models. Although the authors give a short justification of their subsequent sole usage of the KPC model in the Methods, this methodology also requires greater emphasis in the Results and Discussion to make it clear to the reader that the observed findings are confined to one model with the associated limitations.

We now re-emphasize and clarify that the results from Figures 2 and on are from the KPC tumor model in the results and discussion sections. Specifically, we modified the first sentence of the results section of Figure 2: "To determine the meaningful source of LCN2 during pancreatic cancer cachexia, we analyzed LCN2 levels across tissues utilizing the KPC model described in Figure 1" (Lines 172-173 of unmarked manuscript) and added the following sentence to the discussion: "After our initial characterization of LCN2 production across five separate models of cachexia, we focused on the well-characterized KPC model for ensuing genetic and pharmacologic studies" (Lines 572-574 of unmarked manuscript).

2. The authors state that the bone marrow is the predominant site of LCN2 "production" (lines 163 onwards), but I would argue that only LCN2 expression (rather than production) has been assessed. The exact site and timing of LCN2 production during the neutrophil's short half-life appears still unclear, especially as the spleen, liver and bone marrow are all involved in neutrophil clearance, as well as haematopoiesis. Equally, bone marrow LCN2 expression was similar between cachexia and sham. Suggest change "production" to "expression".

This is an excellent point, and while we do demonstrate gene expression data from neutrophils in Figure 2G, we have switched the nomenclature when referring to the bone marrow to “production,” rather than “expression.”

3. The mechanistic MC4R data do not specifically involve the use of LCN2 as a ligand and therefore, although good to note (as LCN2 is reported to be a ligand of MC4R), they do not add significantly to the overall manuscript. The senior author (and other groups) have previously published on the role of MC4R antagonism/inverse agonism as potential therapeutic methods of cachexia amelioration. Do the authors have any novel data on the central administration of LCN2 to mice or the interaction between LCN2 and MCR4?

Similar to other reviewer suggestions, the revised manuscript includes a chronic ICV LCN2 study in both MC4R WT and MC4R KO mice (Supplemental Figure 5H-L). Indeed, we observed a sustained reduction in food intake, body mass, fat mass, and lean mass in MC4R WT mice, but not MC4RKO mice, suggesting central administration of LCN2 regulates appetite in an MC4R-dependent manner.

4. The section on human studies is important but is currently limited in scope and requires expansion. Further patient details are required, including demographics, age, stage etc. How many follow-up assessments were performed, and what treatments were patients receiving (treatment will influence rate and nature of muscle wasting)? Regarding the CT data, what phase of scans were used (they should be consistent; usually portal phase), and why was delta SMI correlated with delta LCN2 (rather than total levels)? Did the authors analyse other CT parameters of wasting (e.g. measures of subcutaneous and visceral fat)?

This is an excellent point, since we cannot account for the influence of cytotoxic chemotherapy or malabsorptive surgeries on systemic LCN2 levels (as in the event of follow up blood draws), we have removed Figure 7A and the associated text. However, we believe the comparisons in Figure 7E-F (which include follow up blood draws for patients) still provide important correlative data, as we are not making a case for a change in LCN2 levels being associated with disease progression, but rather changes in body composition, irrespective of treatment modalities. The revised manuscript includes baseline demographics data concerning the human pancreatic cancer data in Figure 7 (Supplemental table 1 [Figures 7A-C]; Supplemental table 2 [Figures 7E-F]; Supplemental table 3 [Figure 7G]). These table includes basic demographics information, treatment regimens, and staging information. As for phase of CT scan, we (unpublished data) and other have not found the phase of a CT to influence measures skeletal muscle index or visceral fat^{15,16}. The tissue attenuation in muscle and fat are sufficiently non-overlapping that the inclusion of contrast does not influence either the ability to delineate between these two tissue compartments or the accuracy of cross-sectional area. Raw SMI values cannot account for baseline (pre-cancer) normal variation among patients, and the normal values for this are sex-specific. Therefore, we chose to look at the association between loss of muscle mass and LCN2, as delta SMI tells us more about the muscle wasting status of a patient, without being confounded by either baseline muscle mass or patient sex. The revised manuscript now includes new data concerning fat sparing in our mouse model that is significant, therefore we went back and analyzed adipose wasting in our human cohort. Specifically, we observe a significant correlation between rising LCN2 levels and loss of visceral adipose tissue (Figure 7F).

5. The authors highlight the expansion of the neutrophil population and decrease of the lymphocyte population in cachexia. However, discussion of the prognostic importance of the NLR (neutrophil-

lymphocyte ratio) in cancer is limited. What was the NLR of the recruited pancreatic cancer patients? Or other pro-inflammatory measures (e.g. CRP/modified GPS)? And how do they relate to LCN2 levels?

We now expand upon our discussions of NLR in cancer and cancer cachexia in the discussion section (Lines 680-690 of unmarked manuscript). We now include an analysis of NLR and LCN2 levels in Figure 7C, demonstrating a significant positive correlation between the two variables. Notably, we made an error in the initial submission concerning correlative analyses between LCN2 levels and neutrophil/lymphocyte count, as those values were from patients at the time of follow up, not diagnosis. The revised manuscript is now corrected and includes correlative data between LCN2 levels and neutrophils (Figure 7A), lymphocytes (Figure 7B), and NLR (Figure 7C) for patients at the time of diagnosis—no conclusions or level of statistical significance was changed by this. Since our intent is to investigate associations between immune cell profile and LCN2 levels, we believe reporting follow up data concerning LCN2 levels and immune cell count remains valid approach. We now include CA19-9 levels in all supplemental tables. However, correlative analysis between LCN2 levels and CA19-9 were not significant (data not shown).

Minor Comments

Line 108: “muscle-sparing effects of LCN2 blockade is” should be changed to “muscle-sparing effects of LCN2 blockade are”

This text is now corrected.

Line 404: “muscle-sparing effects of LCN2 is” should be changed to “muscle-sparing effects of LCN2 are”

This text is now corrected.

Line 144: “Furthermore, we observe” should be changed to past tense consistent with rest of manuscript

This text is now corrected.

References

1. Zhao P, Elks CM, Stephens JM. The induction of lipocalin-2 protein expression in vivo and in vitro. *The Journal of biological chemistry*. 2014;289(9):5960-5969.
2. Zhang Y, Foncea R, Deis JA, Guo H, Bernlohr DA, Chen X. Lipocalin 2 expression and secretion is highly regulated by metabolic stress, cytokines, and nutrients in adipocytes. *PLoS One*. 2014;9(5):e96997.
3. Xu MJ, Feng D, Wu H, et al. Liver is the major source of elevated serum lipocalin-2 levels after bacterial infection or partial hepatectomy: a critical role for IL-6/STAT3. *Hepatology*. 2015;61(2):692-702.
4. Flint TR, Janowitz T, Connell CM, et al. Tumor-Induced IL-6 Reprograms Host Metabolism to Suppress Anti-tumor Immunity. *Cell metabolism*. 2016;24(5):672-684.
5. Zhu X, Burfeind KG, Michaelis KA, et al. MyD88 signalling is critical in the development of pancreatic cancer cachexia. *J Cachexia Sarcopenia Muscle*. 2019;10(2):378-390.
6. Kjeldsen L, Johnsen AH, Sengeløv H, Borregaard N. Isolation and primary structure of NGAL, a novel protein associated with human neutrophil gelatinase. *J Biol Chem*. 1993;268(14):10425-10432.
7. Bundgaard JR, Sengeløv H, Borregaard N, Kjeldsen L. Molecular cloning and expression of a cDNA encoding NGAL: a lipocalin expressed in human neutrophils. *Biochem Biophys Res Commun*. 1994;202(3):1468-1475.
8. Hingorani SR, Wang L, Multani AS, et al. Trp53R172H and KrasG12D cooperate to promote chromosomal instability and widely metastatic pancreatic ductal adenocarcinoma in mice. *Cancer cell*. 2005;7(5):469-483.
9. Baracos VE, Martin L, Korc M, Guttridge DC, Fearon KCH. Cancer-associated cachexia. *Nature reviews Disease primers*. 2018;4:17105.
10. Fearon K, Strasser F, Anker SD, et al. Definition and classification of cancer cachexia: an international consensus. *The Lancet Oncology*. 2011;12(5):489-495.
11. Michaelis KA, Zhu X, Burfeind KG, et al. Establishment and characterization of a novel murine model of pancreatic cancer cachexia. *J Cachexia Sarcopenia Muscle*. 2017;8(5):824-838.
12. Zhang G, Liu Z, Ding H, et al. Tumor induces muscle wasting in mice through releasing extracellular Hsp70 and Hsp90. *Nat Commun*. 2017;8(1):589.
13. Chen JL, Walton KL, Qian H, et al. Differential Effects of IL6 and Activin A in the Development of Cancer-Associated Cachexia. *Cancer Res*. 2016;76(18):5372-5382.
14. Johnston AJ, Murphy KT, Jenkinson L, et al. Targeting of Fn14 Prevents Cancer-Induced Cachexia and Prolongs Survival. *Cell*. 2015;162(6):1365-1378.
15. van Vugt JLA, Coebergh van den Braak RRJ, Schippers HJW, et al. Contrast-enhancement influences skeletal muscle density, but not skeletal muscle mass, measurements on computed tomography. *Clin Nutr*. 2018;37(5):1707-1714.
16. Rollins KE, Javanmard-Emamghissi H, Awwad A, Macdonald IA, Fearon KCH, Lobo DN. Body composition measurement using computed tomography: Does the phase of the scan matter? *Nutrition (Burbank, Los Angeles County, Calif)*. 2017;41:37-44.

REVIEWER COMMENTS

Reviewer #1 (Remarks to the Author):

The revised manuscript by Olson and co-authors and the rebuttal letter address all of the points raised in my review. The work is topical and well presented and I have no further comments.

Tobias Janowitz

Reviewer #2 (Remarks to the Author):

The authors have significantly improved their manuscript by the addition of experimental data. While the effects of the Lcn2 pathway on skeletal muscle during cachexia remain minimal, new data now also show an effect on adipose tissue mass, thereby expanding the overall concept.

The addition of Lcn2 injections into wt and MCR4 KO mice now supports the general conclusions on the importance of this pathway for central appetite control. However, it remains unclear to the referee whether this experiment was indeed conducted under cancer cachectic conditions, i.e. does Lcn2 aggravate tumor-induced cachexia in wt but not in MCR4 KO mice?

The authors have to clarify this issue to establish a convincing functional case for Lcn2-mediated appetite control in this pathological condition.

Reviewer #3 (Remarks to the Author):

The authors have addressed my concerns and should be congratulated on their impressive work. I only have a couple of minor questions relating to the human patients, plus a few typographical errors:

Abstract Line 44: should read “We demonstrate that...”

Line 180: As per my previous comments, as IHC has been performed, “expression” would be more accurate than “production”

Figure 5: define “BMT” in legend

Line 501: The authors demonstrate that a240 ng/mL plasma LCN2 cutoff stratifies patients by survival outcomes. The authors should specify in the text that this is on univariate analysis only. What happens if multivariate analysis is performed incorporating the variables in Supplemental Table 3? Is plasma LCN2 independently prognostic? And are plasma LCN2 levels dependent on any patient factors e.g. age/sex/stage? Were any of the patients cachectic according to consensus definition, and did LCN predict the presence of cachexia?

Line 770: A short description of the nature of the human controls would be helpful

Supplemental Table 1: Typographic error – FOLFIRINOX

Reviewer #1 (Remarks to the Author):

The revised manuscript by Olson and co-authors and the rebuttal letter address all of the points raised in my review. The work is topical and well presented and I have no further comments.

Thank you for the kind review.

Reviewer #2 (Remarks to the Author):

The authors have significantly improved their manuscript by the addition of experimental data. While the effects of the Lcn2 pathway on skeletal muscle during cachexia remain minimal, new data now also show an effect on adipose tissue mass, thereby expanding the overall concept.

The addition of Lcn2 injections into wt and MCR4 KO mice now supports the general conclusions on the importance of this pathway for central appetite control. However, it remains unclear to the referee whether this experiment was indeed conducted under cancer cachectic conditions, i.e. does Lcn2 aggravate tumor-induced cachexia in wt but not in MCR4 KO mice?

The authors have to clarify this issue to establish a convincing functional case for Lcn2-mediated appetite control in this pathological condition.

Thank you for the kind review. We too agree that the revised manuscript adds substance to the overall pathologic relevance of LCN2 during cachexia. However, we believe that the proposed experiment, in which we would add exogenous LCN2 to WT and MCR4KO cachectic mice, would not further implicate LCN2 in appetite control during pancreatic cancer cachexia beyond the scope of the current manuscript for several reasons. 1) Adding exogenous LCN2 to the brain of an animal with already pathological levels of the molecule present does not represent a feasible biological scenario, 2) for the aforementioned reason, this experiment did not pass approval from our institutional IACUC governance as it was deemed to lack scientific merit, and 3) mice that were administered the melanocortin antagonist AgRP demonstrated a clear role for melanocortin signaling under cachexia conditions (Figure 4). At present, since there is no known LCN2 inhibitor, this is a far more translational and definitive experiment than putting a tumor in an MC4RKO background. We now add to the limitations section of the manuscript describing this issue: "Although deletion of LCN2 improves cachexia-anorexia, direct pharmacologic inhibition of LCN2 in the CNS during cachexia are outstanding, particularly in the context of the putative LCN2-MC4R axis during disease."

Reviewer #3 (Remarks to the Author):

The authors have addressed my concerns and should be congratulated on their impressive work. I only have a couple of minor questions relating to the human patients, plus a few typographical errors:

Thank you for the kind review.

Abstract Line 44: should read "We demonstrate that..."

This text is now corrected.

Line 180: As per my previous comments, as IHC has been performed, “expression” would be more accurate than “production”

This text is now corrected.

Figure 5: define “BMT” in legend

BMT is now defined in the figure legend.

Line 501: The authors demonstrate that a 240 ng/mL plasma LCN2 cutoff stratifies patients by survival outcomes. The authors should specify in the text that this is on univariate analysis only. What happens if multivariate analysis is performed incorporating the variables in Supplemental Table 3? Is plasma LCN2 independently prognostic? And are plasma LCN2 levels dependent on any patient factors e.g. age/sex/stage? Were any of the patients cachectic according to consensus definition, and did LCN predict the presence of cachexia?

We now specify that the survival outcome measures are univariate from univariate analysis alone: (Line 501): “Finally, we identified a 240 ng/mL plasma LCN2 cutoff that stratifies patients by survival outcomes on univariate analysis.” While performing extensive multivariate modeling could be an interesting endeavor, we are limited by several parameters of the present dataset, making a meaningful multivariate model difficult to achieve. Similarly, given the retrospective nature of the dataset, precisely defining cachexia as per the international consensus definition is difficult, since weight loss measurements were inconsistently collected. We hope these data will inform future prospective studies concerning LCN2 levels and cachexia.

Line 770: A short description of the nature of the human controls would be helpful

We now add that these control samples were procured from patients with “no clinical evidence of disease... Specifically, patients that were seen at OHSU with a similar clinical work-up as the pancreatic cancer group, but deemed to have no evidence of pancreatic disease, were included in this study.”

Supplemental Table 1: Typographic error – FOLFIRINOX

This text is now corrected.